# Predictive goal coding by dentate gyrus somatostatin-expressing interneurons in male mice

Mei Yuan [1,2], Aurore Cazala [1], Sven Goedeke [2,3], Christian Leibold [2,3], Jonas-Frederic Sauer [1,4] & Marlene Bartos [1] ✉

To select appropriate behaviour, individuals must rely on encoding of relevant features within their environment in the context of current and past experiences. This function has been linked to goal-associated activity patterns of hippocampal principal cells. Using single-unit recordings from optogenetically identified somatostatin-expressing interneurons (SOMIs) in the dentate gyrus of head-fixed mice trained in a spatial goal-oriented reward-learning task in virtual realities, we show that SOMI activity temporally precedes reward-locations in expert mice characterized by goal-anticipatory behaviour. Predictive goal-encoding by SOMIs is lost after translocation of learned goals to novel previously unrewarded sites leading to rapid reductions in anticipatory behaviour and fast reconfiguration of SOMI activity to times after reward onset in association with reward consumption at novel goal-sites. Chemogenetic silencing of SOMIs caused a loss of memory that trained goal-sites were no longer available. Thus, our data reveal the ability of SOMIs to flexibly encode goal-locations depending on current and past experiences to bias behavioral outcomes.

To make a decision on appropriate behavior, individuals need to update representations of the world from the past with current experience. These representations are computed by extended networks of neurons specialized in the encoding of specific environmental features and are important for successful spatial orientation and navigation[1]. Activity patterns of hippocampal place cells are restricted to discrete locations called place fields and are essential for the encoding of contexts[1]. However, spatially tuned principal cells (PCs) can encode information beyond space[2,3], as indicated by their accumulation at defined spatial features, including goals or cues[4–6]. During physical navigation, place cell populations form map-like representations of the world[3,7], which reconfigure upon changes within that environment[8,9], thereby updating representations by current experiences. GABAergic inhibitory interneurons (INs) innervate large populations of PCs and have been considered to be less universal in

their encoding characteristics than place cells[10], and, thus, to have a narrow role in spatial cognition. However, recent investigations show that hippocampal INs can influence the representation and discrimination of contexts by shaping the spatial tuning of PCs and the spatio-temporal configuration of cognitive maps[6,9,11–14]. Moreover, recent studies in CA1 and the dentate gyrus (DG) indicate that behaviorally relevant non-spatial factors such as visual, auditory, or tactile cues and rewards can drive structured IN activity[14–16], supporting a role for INs in encoding more complex task-related information. However, whether the IN population activity can adapt to context-associated modifications by rapidly adjusting previous track records to current experiences is far from understood.

We trained mice in a goal-oriented learning (GOL) task in which animals had to obtain rewards at fixed locations in a circular virtual reality. We examined the influence of reward-related information on

[1]Institute for Physiology I, Medical Faculty, University of Freiburg, Freiburg, Germany. [2]Faculty of Biology, University of Freiburg, Freiburg, Germany. [3]Bernstein Center Freiburg, University of Freiburg, Freiburg, Germany. [4]Institute of Physiology, Center for Integrative Physiology and Molecular Medicine, Medical Faculty, Saarland University, Homburg, Germany. ✉e-mail: marlene.bartos@physiologie.uni-freiburg.de

activity patterns of SOMIs in the DG in familiar contexts and after dislocation of the goal to novel, previously unrewarded sites. Among the various IN types in the DG, which can be classified based on morphological, physiological, and molecular criteria[17-20], SOMIs target distal apical dendrites of granule cells (GCs), the PCs in this brain area, and other IN types[21,22]. They are, thus, in the optimal position to influence the strength, plasticity, and integration of synaptic inputs from the entorhinal cortex, which transmits information about space, cues, and events to the downstream DG network[23-27], and consequently, the information of task-relevant details of the contexts.

Here, we show that SOMI activity at reward sites depends on the animals' reward expectation in the GOL task. In expert (E) mice, SOMI population activity signaled the expectation of predicted rewards. This predictive signal was rapidly lost if the expected reward was no longer confirmed, pointing to a fast updating of SOMI-mediated signaling. In non-expert (NE) animals, SOMI activity reflected reward consumption. Conclusively, DG SOMIs encode reward-related information depending on the animals' capability to memorize the goal sites. They provide predictive signals in experts as long as the expected outcome is confirmed and mere reward consumption in NEs after the reward is delivered. Relocation of reward sites from trained to novel locations resulted in a rapid reconfiguration of SOMI activity in experts from predictive to consumption signaling. This flexible and fast updating of goal-related memory was lost after chemogenetic SOMI silencing, highlighting the key role of SOMIs in controlling behavioral actions in dependence on current and previously experienced outcomes.

## Results

### Identification of DG SOMIs in behaving mice

To study activity dynamics of DG SOMIs during goal-oriented reward learning, we trained head-fixed mice to run on a spherical treadmill and to collect soymilk at two fixed reward locations on a 4-m-long circular virtual track displayed on monitors surrounding the animals' visual field (Fig. 1a). The virtual reality was characterized by distinct ground patterns and objects along the virtual walls to support the identification of upcoming goal locations on the track (Fig. 1a; "Methods"). Mice were trained to seek rewards in the same virtual reality for 10–15 days (familiar context), and single-unit activity was recorded on two subsequent days.

SOMIs were optogenetically identified (opto-tagged) after bilateral injection of pAAV1:CAG-FLEX-hChR2 (H134R)-mCherry in the dorsal DG of SOM-IRES-Cre mice (Fig. 1a, b). A fraction of 96.8% mCherry$^+$ somata within the hilus were immunopositive for SOM, revealing high specificity in Cre-driven expression of channelrhodopsin-2 (ChR2), and 55.1% of SOM$^+$ cells expressed ChR2 (Fig. 1b, inset). Acutely inserted silicon probe arrays with 4 shanks were connected to an optical fiber to emit blue light for ChR2 excitation and optogenetic SOMI identification (Fig. 1c, d). Two major morphological SOMI subtypes have been previously identified in the DG: Local SOMIs (SOMI$_{local}$) with axonal arborizations in the outer molecular layer of the DG, targeting apical dendrites of GCs and INs[21,28,29], and long-range projecting SOMIs (SOMI$_{proj}$) distributing their axonal fibers in the hilus and projecting to the medial septum and diagonal band of Broca (MSDB) to contact glutamatergic, cholinergic, and GABAergic cells[21]. To differentiate between the two SOMI subtypes during recordings, we implanted a second optical fiber near the fimbria (Supplementary Fig. 1a), with the aim that light application to the hilus will recruit both SOMI subtypes while light delivery to the fimbria will evoke anterogradely propagating action potentials in axons of SOMI$_{proj}$, which can be recorded in the hilus. Histological processing confirmed the correct position of the silicon probes and optical fiber tips (Fig. 1c; Supplementary Fig. 1a, b). Based on light-evoked discharge patterns, four groups of cells could be discriminated: First, 6.9% of the units were classified as SOMIs, as they discharged throughout the entire period of light delivery with a fast on- and offset defined by the light duration (Fig. 1d; Supplementary

Fig. 2a, c). Second, 20.1% of the recorded cells were rapidly silenced upon light onset (Fig. 1d), indicating that they might have been inhibited by presynaptic SOMIs. Third, 4.1% of the neurons showed a delayed and less reliable trial-by-trial recruitment, potentially driven by SOMI-mediated disinhibition of presynaptic SOMIs[21] or other types of GABAergic cells such as parvalbumin-expressing interneurons (PVIs[30]), which in several cases outlasted the time of light delivery (Supplementary Fig. 2b, d). Finally, the largest fraction of units (68.9%) did not respond to the stimulation, defining them as NON-SOM-units within the limits of viral transfection efficacy. Among opto-tagged SOMIs, 71.7% generated action potentials in response to light delivery to both sites the hilus and the fimbria, thereby identifying them as SOMI$_{proj}$, whereas the remaining 28.3% replied solely to hilar stimulation, defining them as putative SOMI$_{local}$ (Supplementary Fig. 2a, f). These data are consistent with previous quantifications of the two SOMI subtypes based on their morphological characteristics[21].

To differentiate opto-tagged SOMIs from other recorded cell types, we examined their spike waveform and bursting properties (Fig. 1e). Putative PCs (813 units; 41.8% of total 1943 units) were characterized by a high bursting index (>3), defined as the average number of spikes in the 3–5 ms bins of the spike autocorrelogram divided by the average spike number in the 200–300 ms bins (Fig. 1f [11,31,32]), and a wide spike waveform with a large trough-to-peak duration (>0.45 ms). In contrast, narrow-waveform INs (404 units; 20.8%) had a low bursting index (<3), and a short trough-to-peak duration (<0.45 ms). Consistent with the strong inhibitory effect of dendritic inhibition on fast-spiking interneurons (FSIs [33]), the activity of a substantial fraction of FSIs was inhibited during SOMI stimulation (148 out of 404 units; Fig. 1e). Wide-waveform cells contained the majority of opto-tagged SOMIs (119 units; 6.1% of all recorded cells; Fig. 1e). As DG SOMIs receive glutamatergic inputs from GCs and mossy cells (MCs [21]), we used cross-correlation analysis of monosynaptically connected units and identified 21 PC-SOMI pairs (latency between putative presynaptic PCs and opto-tagged SOMIs: $2.5 \pm 0.15$ ms; Fig. 1g; "Methods"), with a preference of GCs contacting SOM$_{local}$ and MCs targeting SOM$_{proj}$ (4 out of 5 GC-SOMI$_{local}$, 7 out of 7 MC-SOMI$_{proj}$; 9 PC-SOMI). The mean discharge frequency was significantly higher for FSIs than putative PCs (21.7 Hz vs 2.85 Hz; unpaired $t$-test, $P = 7.2053 \times 10^{-80}$) and opto-tagged SOMIs (Fig. 1h). Light-evoked discharge rates and waveform to burstiness relationships or the temporal link of individual spikes to the phases of local theta (4–12 Hz) and gamma (30–100 Hz) oscillations, did not allow a physiological differentiation between SOMI$_{local}$ and SOMI$_{proj}$ (Supplementary Fig. 3). Taken together, opto-tagging and physiological single unit properties allowed us to reliably identify SOMIs in the hilar area of the DG and to classify them as predominantly wide-waveform non-bursting cells.

### Mice divide into experts and non-experts depending on learning performance

While all mice were trained with the same protocols to perform the GOL task (Methods), we observed marked differences in learning performance based on lick rates and running velocity (Fig. 2). We focused our behavioral analysis on two phases of the task: the pre-reward (defined as the 2 s immediately prior to reward availability) and the post-reward time epoch (2 s immediately following reward delivery; Fig. 2a). Compared to the baseline obtained in a 2 s time window 4–6 s prior to reward delivery, individual E mice were characterized by a significantly increased lick rate in the pre-reward area as they approached the goal site (Fig. 2b, c) and a decelerated running speed (Fig. 2d; "Methods"). In marked contrast, NE animals showed no anticipatory lick rate increase in pre-reward periods (Fig. 2c) and retained a constant average running velocity across the baseline and pre-reward area (Fig. 2d). The mean lick rate further increased in the post-reward zone and was higher in E than in NE mice (Fig. 2c). These differences remained between E and NE mice when lick-rates and

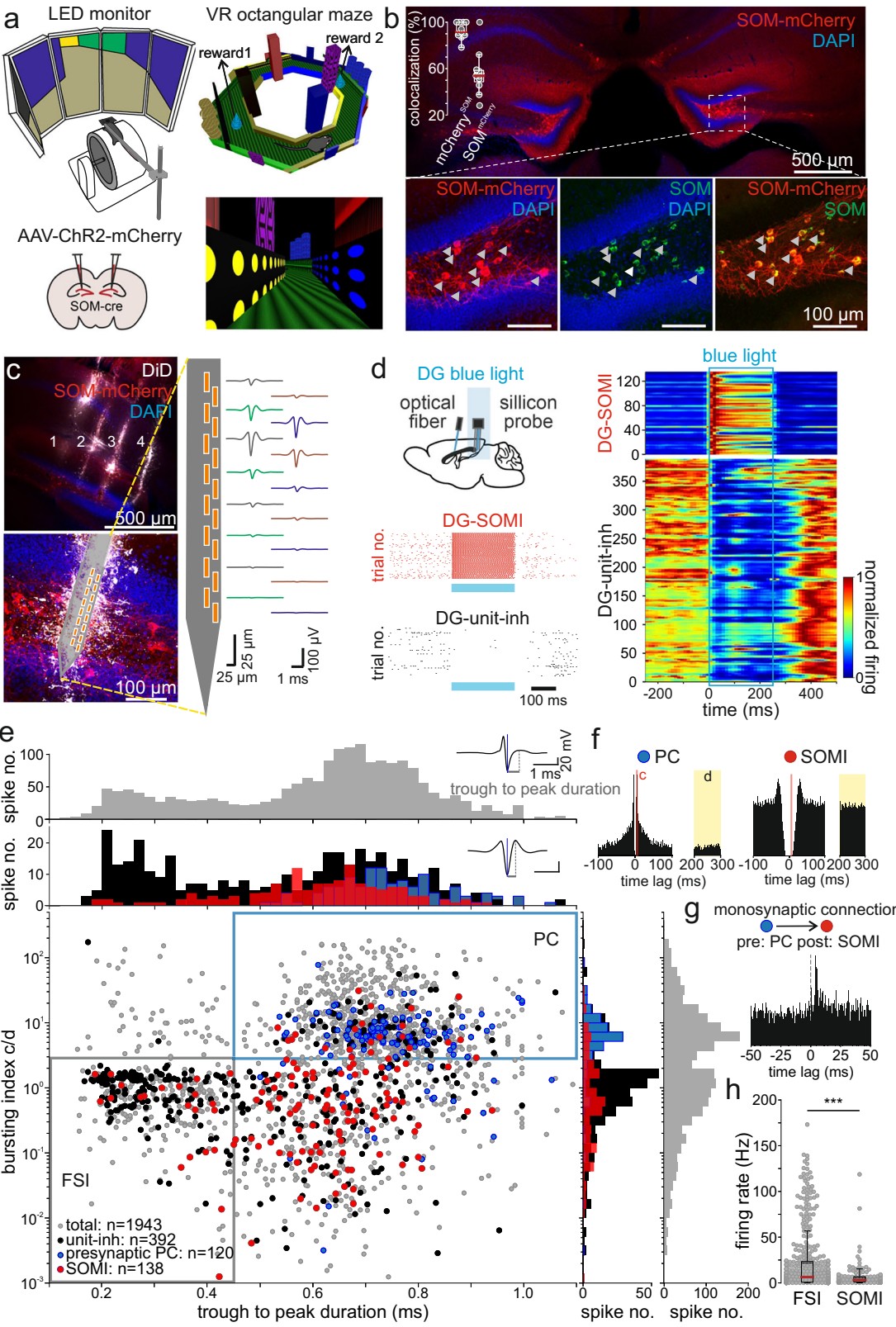

running velocity were plotted against track position rather than time normalized to reward onset (Supplementary Fig. 4a). Consistent with the assumption that anticipation of rewards requires daily learning, anticipatory licking and deceleration in pre-reward zones were not observed after the first, but after ~11 days of training (Supplementary Fig. 4b, c). Moreover, the fraction of laps per daily session in which mice showed anticipatory licking (performance score) monotonically increased across days reaching 74.40 ± 3.1% in E mice after ~11 days of training (Supplementary Fig. 4d–f). Thus, E but not NE mice apparently memorize the task and anticipate rewards at learned goal-locations.

## DG SOMI activity relates to reward expectation in expert mice

Next, we tested whether differences in learning performance are reflected in SOMIs' activity (Fig. 2e–h). In experts, their average

**Fig. 1 | Optogenetic identification of SOMIs during in vivo recordings.**
**a** Schematic of the experimental setup. Head-fixed mice run on a Styrofoam wheel and were trained to collect rewards at defined positions in a virtual circular maze. Bottom left, bilateral injections of rAAV-flex-ChR2-mCherry in dorsal dentate gyri (DGs) of SOM-Cre mice. **b** Coronal section showing expression of ChR2-mCherry in SOMIs in dorsal DGs. Bottom, colocalization of immunohistochemically identified SOM (green) and virally expressed mCherry; blue, DAPI staining; arrows point to somata colocalizing SOM and mCherry. Inset, quantification of viral expression of mCherry in SOM⁺ cells ($n = 3$ mice, $n = 11$ sections of dorsal hippocampus). **c** Left, representative confocal image showing location of silicon probe arrays in the dorsal hippocampus (upper) and the DG (lower; done for all recordings). Right, average waveforms of an example unit depicted along recording sites of one shank. **d** Upper, schematic represents location of the two optical fibers in the fimbria and the dorsal DG and the recording site in the DG. Below, activity of two representative cells during subsequent trials of blue light delivery (488 nm; blue rectangle). Each dot represents an action potential. Right, summary of baseline-normalized activity of all optogenetically identified ChR2-expressing DG SOMIs in response to blue light-on (blue rectangle). DG SOMIs increased their activity ($n = 138$ units, 21 mice, 27 recordings) whereas a large unit fraction was inhibited upon SOMI recruitment

($n = 389$ units, 23 mice, 37 recording sessions). **e** Unit classification based on optogenetic identification, trough-to-peak duration, bursting index, and monosynaptic connections. Each dot corresponds to one unit (1943 units). Optogenetically identified SOMIs are marked as red dots and units inhibited by SOMIs (DG-unit-inh) as black dots. Units with trough-to-peak duration >0.45 ms and bursting index >3 were identified as putative principal cells (PCs; blue area) and those with monosynaptic connections and bimodal distribution of the bursting index (right) as presynaptic PCs (blue dots). Units with trough-to-peak latency <0.45 ms, bursting index <3 represent putative fast-spiking interneurons (FSIs; gray rectangle). Histograms show distribution of trough-to-peak duration (top) and bursting index (right) for the whole recorded unit population (gray bars) and identified cell types (colored bars). Top inset, average spike waveforms of two example units. **f, g** Auto- and cross-correlogram of a monosynaptically connected PC-SOMI example pair. **h** Putative FSIs show significantly higher mean firing rate than optogenetically identified DG SOMIs ($n = 404$ vs 138 units, respectively; unpaired two-sided $t$-test, $P = 4.17 \times 10^{-7}$); each dot represents one cell; boxes, 25th to 75th percentiles; red bar, median; black dashed line, average; whiskers, the largest/smallest data point within the ±1.5 IQR range. Similar data were obtained in 3 cohorts. ***$P < 0.001$.

discharges gradually increased in the pre-reward area to reach a distinct early activity peak shortly before reward onset (median $-0.11 \pm 0.11$ s; 37 cells, 11 mice; Fig. 2f), and superimposed with the maximal rate of licks (Fig. 2e, bottom). Subsequently, SOMI activity rapidly declined towards lower values in the post-reward zone before reward consumption ceased (Fig. 2e). Despite these average SOMI activity dynamics, a minor fraction of SOMIs (29%) reduced their activity at pre-reward sites (Fig. 2e). Importantly, the mean SOMI firing rate was already significantly increased above baseline during the pre-reward period (Fig. 2h). In marked contrast, in NEs, SOMI activity was less temporally defined and showed on average a significant increase after reward onset (median $1.21 \pm 0.31$ s; eight cells, six mice; Fig. 2f), coinciding with the main phase of reward consumption (Fig. 2e). Indeed, SOMI firing rates in NE mice were not affected by the entry into pre-reward zones and increased only in post-reward areas (Fig. 2h). Comparable results were obtained when SOMI activity was plotted against track distance (Supplementary Fig. 4a), and were independent of the SOMI subtype (Supplementary Fig. 5a, b). Despite these clear differences in SOMI dynamics between experts and NEs, their mean discharge frequency was similar during baseline (Fig. 2h). Noticeably, the fraction of reward-anticipatory SOMIs was substantially larger in E than in NE animals (75% vs 40.9%, respectively; Fig. 2g). To disentangle the potential contribution of the key behavioral variables licking, speed and reward delivery to SOMIs' activity, we applied a general linear model (GLM, "Methods"). The model revealed that the activity for a larger proportion of SOMIs could be significantly explained by the parameters in E as compared to NE animals. (Fig. 2i). To further dissect the impact of the variables on SOMI activity in experts, we constructed reduced models in which one of the parameters was randomly permuted. This analysis revealed a significant effect of licking and reward but not of speed or acceleration (Supplementary Fig. 5c). To further assess whether SOMI activity can be explained during phases of reward anticipation, we restricted the same analysis for a time window of 3 s before reward onset. Similarly, only licking but not speed or acceleration significantly affected SOMI activity in experts during the reward anticipatory phase (Supplementary Fig. 5d). Thus, our data show that in experts SOMI activity at goal-locations is modulated by anticipatory behavior.

To directly address whether SOMI activity patterns bear information on anticipated reward locations, we applied the inverse approach and used differences between baseline activity (time before pre-reward area) and the SOMI activity recorded in intervals of varying lengths starting 1.5 s prior to reward onset (Supplementary Fig. 6) to predict rewards, by using a cross-validated maximum likelihood (ML) decoder (see section "Methods"). Significant reward prediction was

achieved on the single cell level in 30% of the recorded cells (>1 spike/1.5 s on average) in E animals before reward onset but for none of the cells in NE mice as no significant rate changes were observed during the pre-reward period (Supplementary Fig. 6a). Extending the ML decoder to the population level, we found that only 10 SOMIs were necessary to predict rewards with 80% accuracy in experts based on a 1.5 s period before reward onset (Supplementary Fig. 6b; "Methods"). This criterion was not reached in NEs even if we included all recorded SOMIs (Supplementary Fig. 6b). Repeating the same analysis with activity traces from which speed and acceleration effects were subtracted according to GLM fits revealed successful decoding at 75% prediction performance (Supplementary Fig. 6c).

To assess whether DG SOMIs respond to sensory stimuli in our GOL task we pseudo-randomly applied rewards (RRs), light-pulses, tactile stimuli, and tones (Supplementary Fig. 7a–c). A large fraction of SOMIs was recruited by RRs and air-puffs (38.9% and 36.1%, respectively), only few by light-pulses (16.7%), and none by tones (Supplementary Fig. 7c). This finding was independent of the morphological SOMI subtype (Supplementary Fig. 7d, e). Notably, the fraction of SOMIs raising their discharge rate during pseudo-random foraging was similar to the fraction of NE SOMIs obtaining rewards at fixed reward locations but substantially lower than the one of expert SOMIs (RR 38.9% vs NE 40.9% vs E 75%; Fig. 2g). Moreover, SOMI firing reached its peak after RR onset (median $0.9 \pm 0.15$ s; Supplementary Fig. 7c), similar to NE SOMIs at fixed goal-locations (median $1.21 \pm 0.31$ s; Fig. 2f). Thus, a substantial fraction of DG SOMIs exhibit discrete reward-related discharges in the absence of training, which fits to SOMI responses observed in CA1[15].

## Reward-predicting SOMI activity is not explained by running velocity or acceleration

Expert performance in the spatial GOL-task coincided with a reduction in running speed in the reward zone to $5.19 \pm 0.68$ cm s⁻¹. We therefore tested whether reward-anticipatory firing rate changes of SOMIs could be explained by speed modulation (Fig. 3a, b). SOMIs altered their activity in dependence on the transition from immobility to running, defined as speed change from 0 to $\geq 3$ cm s⁻¹ in $\leq 3$ s (Fig. 3c). One fraction of SOMIs (45.4%) showed highest discharges during immobility, with a rapid decline in firing preceding running onset (OFF-SOMIs; Fig. 3c). A second fraction of cells (20%) displayed the opposite, lowest neuronal activity during immobility with a rapid rise preceding running onset (ON-SOMIs; Fig. 3d). The activity of the remaining SOMIs (34.6%) did not vary with rest-running transitions (NON-SOMIs; Fig. 3e; Supplementary Fig. 8a, b). The activity of OFF- and ON-SOMIs was similar when averaged across the entire track and all laps (Fig. 3f).

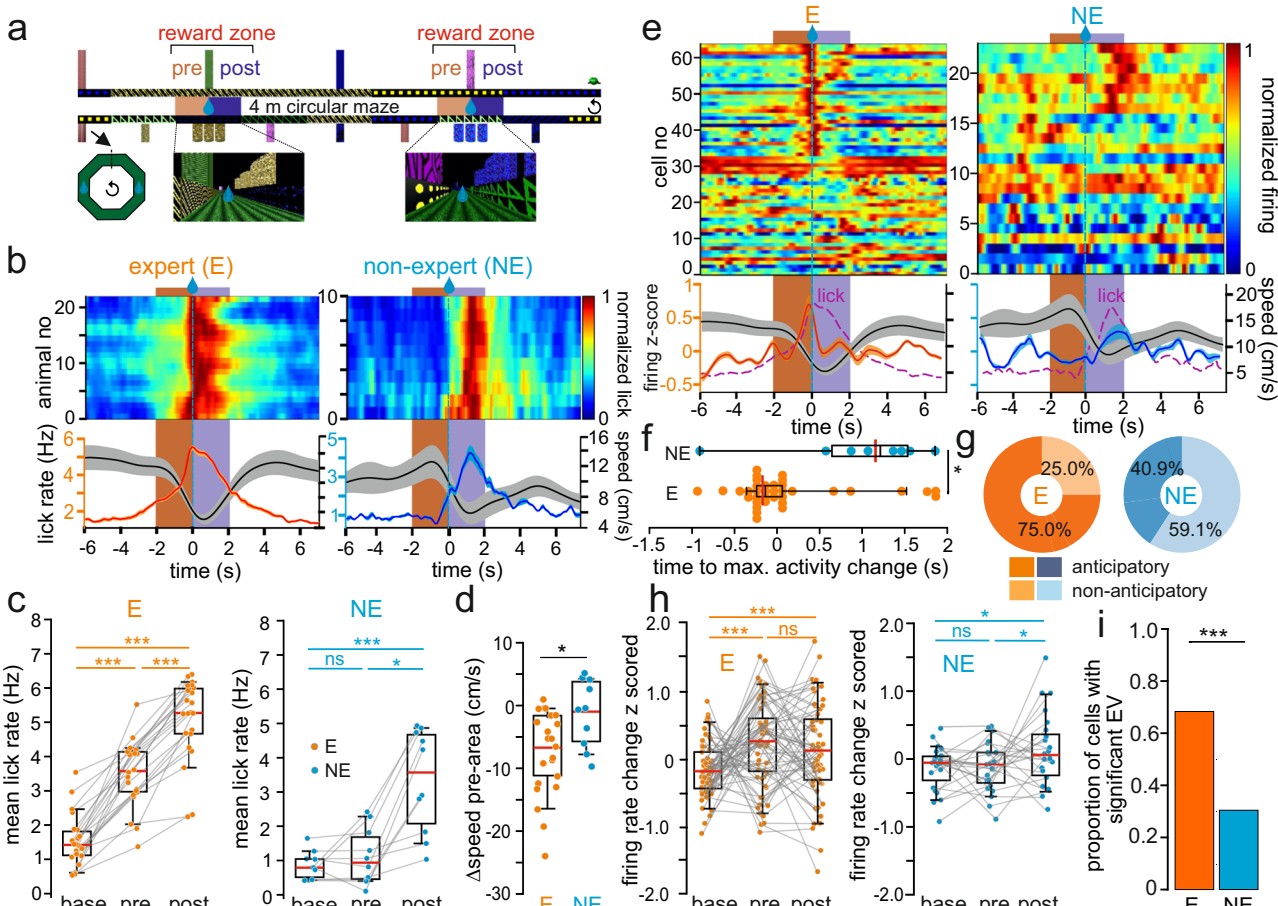

**Fig. 2 | Reward expectation-dependent SOMI responses at goal-locations.**
**a** Upper, schematic of the maze was stretched to visualize reward zones defined as the 2 s prior to reward availability as pre- (brown) and the 2 s after reward onset as post-reward area (blue). Bottom, wall patterns, and virtual objects at pre-reward zones. **b** Classification of mice in the goal-oriented learning task based on lick behavior in experts (E) showing increased licking in the pre-reward area as compared to the remaining track (ANOVA with repeated measure and post-hoc Bonferroni's correction, $n = 16$ mice), and non-experts (NE) lacking increase in lick rate in pre-reward area ($n = 7$ mice). Top, normalized lick rate for each recording aligned to onset of reward delivery (blue striped line). Bottom, average lick rate (orange line) and mean running speed (black line). E animals increase lick rate in the pre-reward area. Values, averages over 100 ms bins, smoothed with a Gaussian filter ($\sigma = 1$ bin) over time. **c** Lick rate of E mice increases in pre-area as compared to remaining track (base, $t = 10.65$, $P = 10^{-10}$) and remains elevated after reward delivery (post, $t = 16.68$, $P = 10^{-13}$). NE mice increase lick rates after reward availability (pre vs. base: $t = 1.07$, $P = 0.31$, post vs. base: $t = 5.40$, $P = 0.0004$). One-way repeated measures ANOVAs followed by paired $t$-tests with Sidak correction ($n = 22$ E and 10 NE sessions). Baseline activity in E and NE mice was not different (two-sided Wilcoxon rank sum-test, $P = 0.4284$). **d** Change in running speed in pre-reward area as compared to remaining track in E and NE mice (unpaired two-sided

$t$-test, $t = -2.68$, $P = 0.012$; $n = 22$ E vs 10 NE sessions). **e** Activity of all SOMIs near reward zone in E and NE mice (E, $n = 64$ cells, 11 mice; NE, $n = 22$ cells, 6 mice). Top, SOMI firing normalized to its peak activity. Bottom, z-scored SOMI activity superimposed with average running speed (black) and lick rate (purple; same as in (**b**)). **f** Time to maximal activity change for reward-modulated cells in E and NE mice ($n = 37$ vs 8 cells; see "Methods") relative to reward onset (Wilcoxon Rank-Sum test, $P = 0.0126$). **g** Fraction of SOMIs showing anticipatory or non-anticipatory activity in E and NE. **h** E SOMIs increase firing rate in pre-area compared to baseline ($t = 3.47$, $P = 0.0009$; post vs. base: $t = 2.52$, $P = 0.014$, $n = 64$ cells). NE SOMIs do not show changes in average firing rates ($F = 1.99$, $P = 0.15$, $n = 22$ cells). One-way repeated measures ANOVAs followed by paired $t$-tests with Sidak correction. **i** Proportion of SOMIs with significant explained variance of spike rates at reward zones for E and NE mice using a GLM with behavioral predictors including lick rate, running speed and reward location (two-sided Chi-square test, $P = 0.0018$, Chi-square 9.792; $n = 60$ E and 20 NE SOMIs); ***$P < 0.001$, *$P < 0.05$; ns not significant. Lines with shadows represent mean ± SEM; circles represent individual sessions (**c**, **d**) or cells (**f**–**h**). Similar data were obtained in three animal cohorts. Boxes represent 25th–75th percentiles; red bar, median; black line in (**f**), mean; whiskers, the largest/smallest data point within the ±1.5 IQR range.

Despite these transient activity changes, a substantial fraction of OFF-, ON- and NON-SOMIs showed neither significant speed-modulation during phases of steady-state running (82%, 69.2%, 89%, respectively; Supplementary Fig. 9), nor to acceleration (Supplementary Fig. 10). The division of SOMIs into OFF-, ON- and NON-cells was neither related to their morphological subtypes (Supplementary Fig. 8c, d), nor to the recording session or the design of the virtual environment (Supplementary Fig. 11). However, OFF-SOMIs formed the largest fraction of cells among both SOMI$_{local}$ and SOMI$_{proj}$ (~50%; Supplementary Fig. 8d), suggesting that a locomotion-on and -stop signal might be of particular relevance for DGs' network computations and its target area the MSDB, which controls upcoming starts of locomotion[34]. As E mice

reduced their mean running velocity to -5 cm s$^{-1}$ in the reward zone (Fig. 2b), which is above the transient speed modulation of OFF- and ON-cells, we infer that reward area-related SOMI activity dynamics do not relate to rest-running transitions.

## Reward translocation affects SOMI activity dynamics in expert mice

If a rise in SOMI activity in the pre-reward area conveys information on the expected outcome at learned goal-locations in our task, then shifting reward sites from memorized familiar (FAM) to novel previously unrewarded (NOV) locations should influence anticipatory behavior and SOMI activity dynamics in E mice (Fig. 4). Indeed, pre-

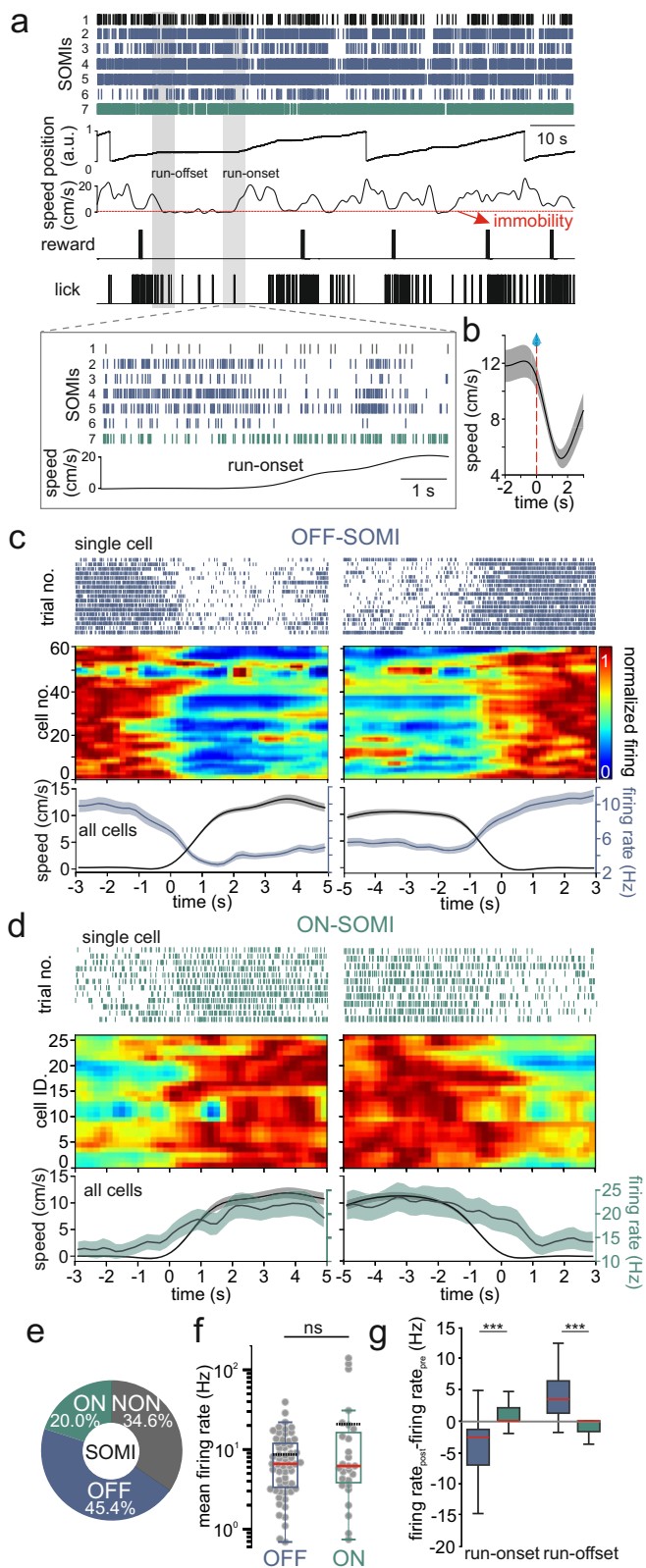

**Fig. 3 | Activity of DG SOMIs is modulated during transitions between locomotion and resting. a** Single unit activities of optogenetically identified DG SOMIs relative to running speed and licks during 2 laps on the maze. Red horizontal line refers to threshold between immobility and locomotion (≥0.5 cm s⁻¹). Inset, expanded view on activities of DG SOMIs during locomotion onset (gray shaded time window). **b** Average running speed of all recorded mice and sessions near rewards. Red dashed line refers to the time of reward onset. **c** SOMI activity during transitions between immobility and running. SOMIs, which showed decreased discharges during running onset (left) and the opposite activity change during running offset (right), were defined as OFF-SOMIs (blue units). Upper, OFF-SOMI activity during individual transitions between running and immobility. Bottom, activity heat map for individual OFF-SOMIs and their averaged discharge rate as well as velocity during initiation (left) and termination of running (right; $n = 61$ cells, 17 mice, 19 recordings). Single-cell activity was normalized to maximal mean firing of the cell. Data were aligned to running on- or offset. **d** Same as in (**c**) but for ON-SOMIs (green units) characterized by increased activity during locomotion onset and reduced discharge frequency during running offsets ($n = 27$ cells, 11 mice, 14 sessions). **e** Fraction of OFF-, ON-SOMIs and SOMIs, which were not modulated during transitions between locomotion and immobility (NON-SOMIs, black units; see also Supplementary Figs. 8–10). **f** Mean firing rates of OFF- and ON-SOMIs during the entire recording session ($n = 61$ and 26 cells, respectively; two-sided Wilcoxon rank-sum test, $P = 0.65$). **g** Mean activity difference between immobility and locomotion during running onsets (left, paired two-tailed $t$-test, $P = 1.16 \times 10^{-8}$; $n = 59$ cells, 16 mice) and offsets (right, $P = 1.00 \times 10^{-9}$; $n = 27$ cells, 11 mice). Boxes in (**f**, **g**) represent 25th–75th percentiles; red bar, median; black dashed line, mean; whiskers, the largest/smallest data point within the ±1.5 IQR range. Areas in (**b**–**d**) with lines represent mean ± SEM. ***$P < 0.001$; ns not significant. Similar data were obtained in three mouse cohorts. For details, see "Methods".

maximal SOMI activity was significantly delayed in experts to times after reward onset in NOV zones (Fig. 4d, e) and anticipatory SOMI activity was abolished in ORI areas when averaged across all laps (Fig. 4d, blue). However, as E mice rapidly learned that the originally trained goal-sites are no longer rewarded, we quantified SOMI activity during the initial 4 runs at ORI areas (ORI-4) after translocation. Indeed, SOMI activity was significantly increased in the ORI-4 pre-reward zone and dropped to baseline values at the beginning of post-reward sites (Fig. 4d, f, pink), indicating that SOMIs rapidly reconfigure their activity if expected rewards were not confirmed.

To further prove that SOMI firing rapidly changed in ORI-zones after reward translocation from FAM to NOV goal-sites, we analyzed activity correlations of SOMIs between the last 5 laps in the FAM reward area prior to translocation and the first 4–5 laps in the ORI areas just after goal translocation. The mean activity correlation was significantly higher than between the last 5 laps in the FAM and the last 5 laps in ORI goal-areas (Supplementary Fig. 12b), indicating that SOMI activity rapidly changed across subsequent laps, as the animal experienced that the trained goal-sites are no longer rewarded. Thus, SOMIs provide predictive encoding of expected outcomes, which is rapidly lost once the outcome is not confirmed.

In marked contrast to SOMIs, mean activity shifts following reward translocation were not observed in FSIs (Supplementary Fig. 13). In fact, their mean discharges decreased in experts and NEs as the mice approached the reward locations, reaching a minimum in the post-reward area (Supplementary Fig. 13). Consistent with the primary positive speed modulation of FS PVIs in the DG[12], the time course of activity changes of FS interneurons in the FAM and NOV reward zones closely reflected the animals' running speed with the mean discharge reaching its minimum slightly earlier in E than in NE animals (0.57 vs 0.7 ms, $P = 0.036$, Mann Whitney Rank Sum test; Supplementary Fig. 13c, bottom). Thus, predictive goal encoding involves SOMI but not FSIs.

## DG SOMI activity is required for flexible goal learning
If DG SOMIs provide predictive signaling as long as the expected outcome is confirmed, then silencing them in experts should influence

reward licking was absent in NOV reward zones (Fig. 4a, b). Importantly, E mice displayed markedly reduced but still significant anticipatory licking as well as reduced running velocity at originally trained (ORI) goal-areas (Fig. 4a, b), indicating that they memorized previously rewarded goal-sites. Similar results were obtained when behavioral parameters were plotted as a function of track position instead of time (Supplementary Fig. 12a). Related to these behavioral changes,

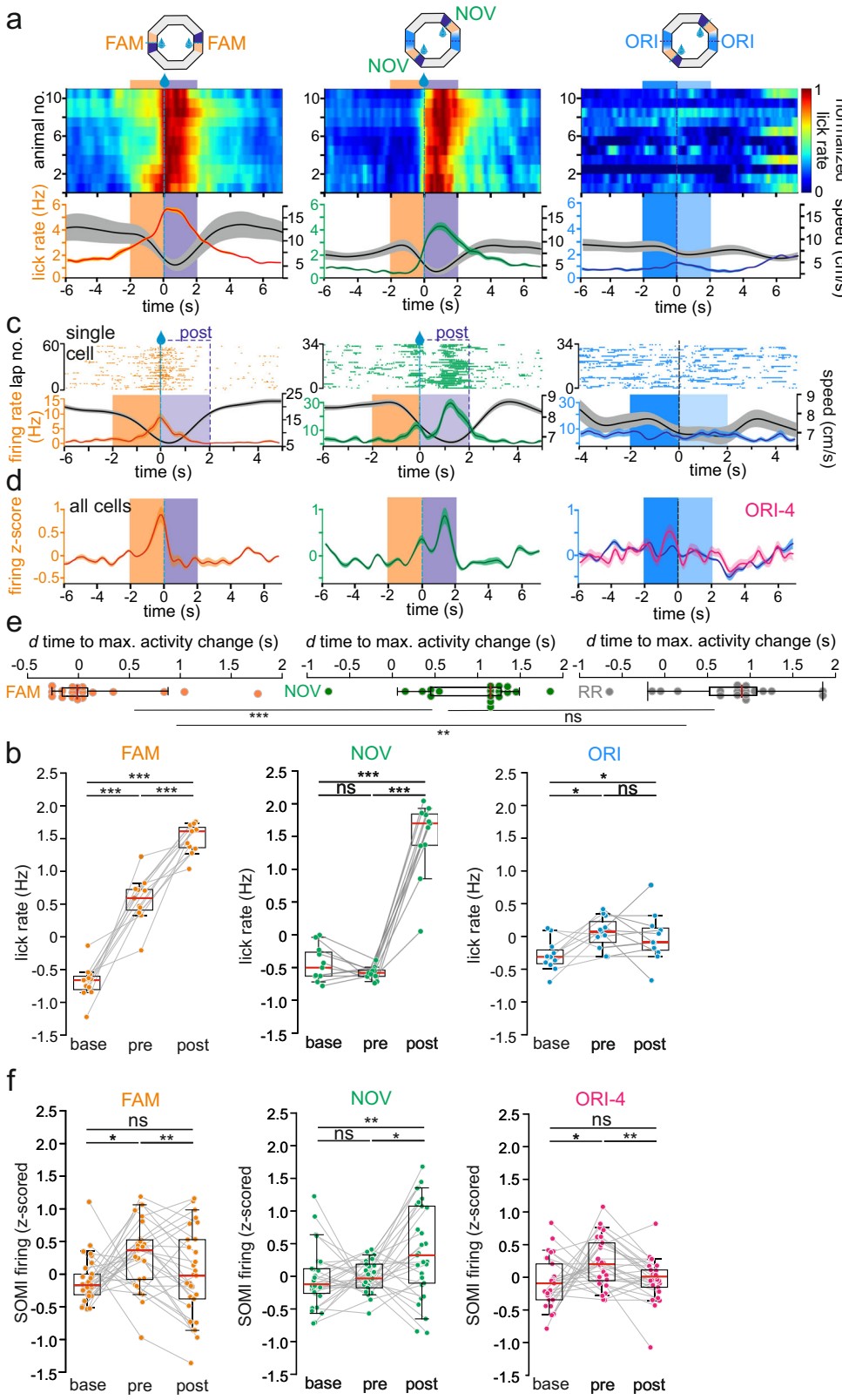

the animals' ability to flexibly alter its performance at the originally trained reward location after goal translocation. To address this hypothesis, we expressed AAV-FLEX-hM4D(Gi)-mCherry bilaterally in the dorsal DG of SOM-Cre mice to inhibit SOMIs upon intraperitoneal injection of the hM4Di agonist clozapine (Fig. 5; "Methods"). Control SOM-Cre animals expressed AAV-FLEX-hSyn-mCherry and underwent similar clozapine injections. We recently demonstrated the high

specificity in the viral expression of hM4Di and mCherry in SOM-Cre mice and in the marked reduction of in vivo DG SOMI activity using similar procedures[14]. Following training in the GOL task to expert levels, reward sites were translocated to NOV locations and running as well as lick behavior were quantified on 2 consecutive days (Fig. 5a). Intriguingly, both groups of mice reduced their mean running speed in the pre-reward ORI zones in anticipation of the memorized reward

**Fig. 4 | Reward translocation affects reward expectation-related SOMI activity in E mice. a** Top, schematic representation of reward-translocation experiments. The virtual circular maze remained identical except for the translocation of reward sites. Color code for familiar (FAM, orange), novel (NOV, green) translocated reward sites and the original familiar reward areas (ORIs, blue). Bottom, summary of lick-rate and running velocity for individual animals ($n = 11$ mice). Note diminished anticipatory licking in NOV pre-reward area (middle) after reward translocation and residual licking in ORI area. **b** Lick rates increased prior to reward availability at FAM sites (pre vs. baseline: $t = 8.63$, $P = 10^{-6}$, post vs. baseline: $t = 20.47$, $P = 10^{-9}$, pre vs. post: $t = -6.42$, $P = 10^{-5}$), after reward delivery at NOV sites (pre vs. baseline: $t = 1.68$, $P = 0.12$, post vs. baseline: $t = 10.87$, $P = 10^{-7}$, pre vs. post: $t = -11.6$, $P = 10^{-7}$) and residual anticipatory licking prior to expected reward delivery at ORI sites (pre vs. baseline: $t = 3.26$, $P = 0.009$, post vs. baseline: $t = 1.80$, $P = 0.10$, pre vs. post: $t = 0.64$, $P = 0.53$, $n = 11$ cells; one-way repeated measures ANOVAs followed by paired $t$-tests with Sidak correction). **c** Upper, raster plot of firing activity of a representative SOMI on subsequent laps in the FAM, NOV, and ORI area after reward translocation. Lower red line depicts the mean activity of the same SOMI; the black line shows the mean running velocity of the respective animal. **d** Same as (**c**) but for all recorded SOMIs ($n = 29$ cells, 5 mice) in the FAM, NOV, and ORI area and for all laps. Right, the pink line denotes mean SOMI activity during the first 4 runs in the ORI area (ORI-4), the blue line for the remaining 11 laps. Note

SOMIs rapidly modulate their activity in the NOV and ORI zone after reward-translocation. **e** Summary of delays between reward onset and the maximal SOMI activity change in E mice in the FAM, NOV reward area and upon random reward (RR) delivery (one Way ANOVA repeated measure, $n = 18$ cells, 5 mice, $P = 0.00084$; followed by paired two-sided $t$-test with Sidak correction FAM $-0.01 \pm 0.13$ vs NOV $1.15 \pm 0.15$ s $P = 0.001$, FAM vs RR $0.9 \pm 0.15$ s $P = 0.007$, NOV vs RR $P = 0.473$). **f** SOMIs significantly elevate their firing rate at FAM pre-reward sites (pre vs. baseline: $W = 100$, $P = 0.01$, post vs. baseline: $W = 184$, $P = 0.481$, pre vs. post: $t = 2.84$, $P = 0.008$; $n = 29$ cells, 5 mice). SOMI firing rate increase at NOV sites after reward delivery (pre vs. baseline: $t = 0.36$, $P = 0.718$, post vs. baseline: $t = 2.76$, $P = 0.01$, pre vs. post: $t = -2.97$, $P = 0.006$). During the initial four runs (ORI-4), SOMIs showed residual anticipatory firing rate increases prior to expected reward delivery (pre vs. baseline: $t = -2.49$, $P = 0.019$ ($P_{critical} = 0.017$), post vs. baseline: $t = 0.51$, $P = 0.615$, pre vs. post: $t = 3.66$, $P = 0.001$, $n = 29$ cells). One-way repeated measures ANOVA followed by two-sided Wilcoxon signed-rank or paired t-tests with Sidak correction. Dots in (**c**) represent individual spikes, circles in (**b**, **e**, **f**) individual cells. Lines with shadows represent mean ± SEM. Circles connected by lines represent one measured cell. Boxes represent 25th–75th percentiles; red line, median; whiskers, the largest/smallest data point within the ±1.5 IQR range. Similar data were obtained in three animal cohorts. *$P < 0.05$; **$P < 0.01$; ***$P < 0.001$; ns not significant.

locations, however, this reduction was markedly smaller in controls as compared to hM4Di-mice already on day 1 (Fig. 5b). Moreover, the mean lick rate markedly dropped from pre- to post-reward areas in controls on day 1, remained low on the next day, but stayed high in hM4Di animals until it reached similar baseline lick rates as controls ($P = 0.3713$, $t$-test; Fig. 5e). Consistent with the fast learning of experts that ORI sites are no longer rewarded, control mice reduced their lick rate across laps (Fig. 5c–e, upper), further indicating that clozapine had no effect on the animals' ability to seek rewards and to learn, but remained high in hM4Di-mice on consecutive runs through the virtual reality (Fig. 5c–e, lower). Thus, our data demonstrate that DG SOMI activity is important for fast updating goal-memory and flexibly adjusting food-seeking behavior in relation to current experiences.

## Discussion

Here, we combined electrophysiological, chemogenetic, and behavioral studies in mice to unravel the activity profiles of optogenetically identified SOMIs during goal-oriented reward learning in virtual realities. Our results provide seminal insights on the information content of active SOMIs, which is forwarded to the local DG circuitry and to the downstream MSDB to support task-related behavioral actions. We show that DG SOMIs but not FS-interneurons reconfigure their activity patterns in dependence of the animal's performance (Figs. 2 and 4). In E mice, which memorized their virtual surroundings, SOMI activity encoded the anticipated rewards at trained goal-sites. The "reward-prediction" signal, characterized as increasing SOMI activity before the onset of the expected reward was rapidly lost once the reward was no longer confirmed and reconfigured to a "reward-consumption" signal, characterized by elevated SOMI activity after reward onset, at novel previously unrewarded locations (Fig. 4). In marked contrast, in NE mice characterized by missing reward-anticipatory behavior, predictive signaling was lacking and SOMI activity encoded pure reward consumption (Fig. 2). Chemogenetic silencing of SOMIs in the dorsal DG caused reduced flexibility in learning and adjustment of the behavior to novel goal-locations in an otherwise familiar environment. These data indicate that SOMIs play a crucial role in updating memory based on relevant changes within the animals' environment and reconciling current with past experiences to decide on appropriate actions.

Although advances in transgenic techniques have provided support for identifying DG SOMIs, it is still a challenge to disambiguate their subtypes due to a lack of molecular profiles. Given the morphological subdivision of DG SOMIs in at least two subtypes[21], we

employed optogenetic stimulation of ChR2-expressing SOMIs locally in the DG as well as remotely in the fimbria to evoke anterogradely propagating action potentials and to thereby differentiate SOMI$_{local}$ from SOMI$_{proj}$. Both subtypes showed similar activity changes during GOL-tasks in both E and NE mice. This similarity is unexpected as the two subtypes receive afferent excitatory inputs originating from different sources. SOMI$_{local}$'s appear to be predominantly targeted by GCs whereas SOMI$_{proj}$'s from hilar MCs[21,35]. Future investigations will unravel the connectivity of SOMI subtypes within the DG and the strength of glutamatergic inputs provided by the upstream entorhinal cortex and by the downstream back-projecting CA3 PCs[14,36]. Although we cannot exclude the possibility of unequal opsin expression and thereby the potential underestimation of the fraction of SOM$_{local}$, in 76.9% of the cases we simultaneously recorded optogenetically identified local and long-range projecting SOMIs in one animal. We are therefore confident that the activity of both subtypes in one animal is modulated by rewards, by transitions between immobility and locomotion, as well as by vision and touch (air-puff). We expect that future transcriptome analysis combined with viral approaches will allow specific manipulation of SOMI subtypes and thereby shed light on their potential different functional integration in the DG-MSDB network, their role in defining GC population activity encoding environmental features[14,37,38], and in influencing locomotion in GOL-tasks[39,40]. Although the majority of SOMIs in experts increased their discharges, some cells reduced or did not alter their activity (Fig. 2e). This diversity is very likely caused by a network effect, as SOMIs are strongly interconnected with other GABAergic neurons and PCs[30].

Our study complements previous work in CA1, indicating that SOMI activity is reward-modulated in GOL-tasks[16,41,42]. However, it adds key insights in the complexity of SOMI signaling by demonstrating that their reward-related activity is dichotomous between high and low performing mice and rapidly reconfigures after goal-translocation. The predictive vs. consumption-driven SOMI signaling suggests that experts and NEs may follow different navigational strategies in accordance to their commitment in the task. Indeed, active engagement in spatial exploration has been shown to be essential for maintaining cognitive hippocampal maps[43]. Reward-seeking E mice seem to predominantly rely on memorized cues, which provide information on predicted goal-locations in the familiarized context, whereas, low-performing reward-consuming NE animals seem to rely on afferent reward-related stimuli such as smell and taste. The notion that the animals' reward seeking behavior relates to temporally defined SOMI activity, is supported by our finding that the fraction of predictively

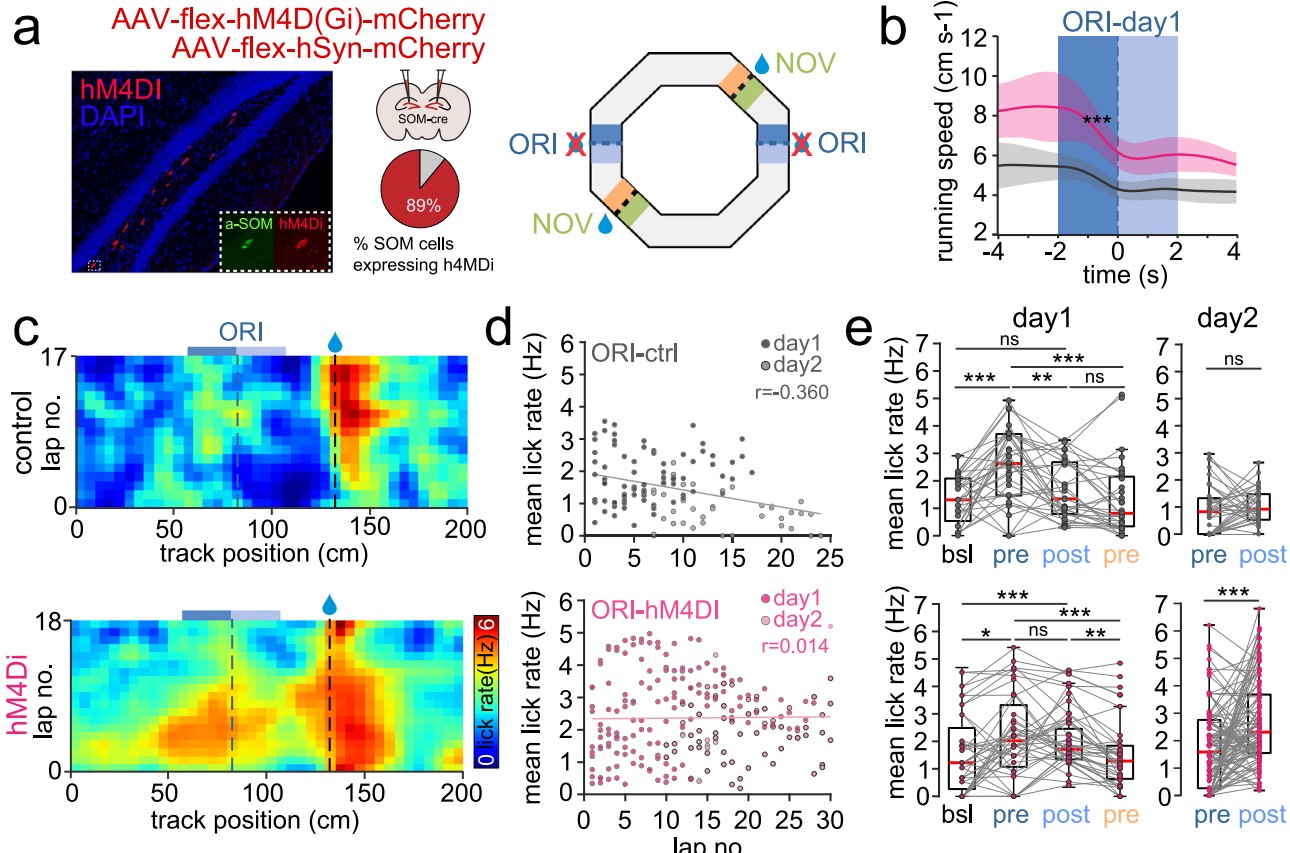

**Fig. 5 | Reduced flexibility in updating reward-seeking behavior under conditions of silenced SOMIs in the DG. a** Left, confocal image stack of hM4Di expression in the DG hilus. Inset, antibody labeling against SOM (green) of an hM4Di-expressing cell (red). Middle, schematic representation of AAV-flex-hM4Di-mCherry or AAV-flex-hSyn-mCherry (control) bilateral injections in the dorsal DG of SOM-Cre mice (*n* = 4 and 5, respectively). Fraction of immunohistochemically labeled SOMIs coexpressing hM4Di. Right, schematic of the experimental set-up with the original trained reward zone (ORI) and the novel (NOV) reward area after reward translocation. **b** Running speed of control (black) and hM4Di-expressing (pink) SOMI mice after clozapine injection in the ORI area after translocation on day 1. Data are aligned to the original reward site (striped line). Linear regressions fitted to speed reduction in controls and hM4Di mice in pre-reward zone (*n* = 4 and 5 mice, respectively; two animal cohorts). Two-sided Mann–Whitney rank sum test, $P = 3.66 \times 10^{-4}$. **c** Lick rate on subsequent laps (upper) from a control and hM4Di-mouse upon clozapine injection. Dark and bright blue areas indicate pre- and post-reward ORI zones, respectively. Blue and black striped lines indicate the onset of the original and the novel reward, respectively. **d** Lick rate during subsequent laps plotted against lap number for control (gray, upper) and hM4Di-mice (pink, lower)

on day 1/2. Data were fit to a least-squares linear regression to assess the correlation between lick rate and lap number as defined by Pearson *r*. Control $P = 5.74 \times 10^{-5}$, hM4Di $P = 0.84$. **e** Bar graphs summarize mean lick rates for baseline (bsl), pre- and the post-reward ORI areas, and pre-reward NOV areas for recording day 1 (left) during the first 4 runs, and pre- and the post-reward ORI for recording day 2 (right) for all runs. Lick rates decline in controls across days but stay elevated in hM4Di mice (controls from 32 ORI-4 and 37 ORI measures in *n* = 4 mice on day 1 and 2, respectively; two-way repeated measure ANOVA tukey posthoc; bsl vs pre paired *t*-test, $P = 2.2963 \times 10^{-5}$, bsl vs post $P = 0.1606$, pre vs post $P = 0.0045$, pre vs NOVpre $P = 0.0003$, post vs NOVpre $P = 0.1442$ on day 1; $P = 0.5277$, day 2; hM4Di mean lick rates from 40 ORI-4 and 106 ORI measures from 5 mice on day 1 and 2, respectively; paired *t*-test, bsl vs pre $P = 0.0116$, bsl vs post $P = 8.7619 \times 10^{-5}$, pre vs post $P = 0.9909$, pre vs NOVpre $P = 0.0342$, post vs NOVpre $P = 3.723 \times 10^{-6}$ on day 1; $P = 3.17 \times 10^{-4}$, day 2). Lines with shadows depict mean ± SEM; circles represent mean values for individual ORIs; boxes represent 25th–75th percentiles; red line, median; whiskers, the largest/smallest data point within the ±1.5 IQR range *$P < 0.05$; **$P < 0.01$; ***$P < 0.001$; ns not significant.

active SOMIs was markedly higher in E than NE mice (Fig. 2g). More-over, predictive encoding was rapidly lost once the expected outcome could no longer be confirmed (Fig. 4). Finally, chemogenetic SOMI silencing caused a loss in flexible updating of goal-related memory as reflected by the animals' inability to learn that the original goal-sites are no longer rewarded (Fig. 5). Taken together, our data extend upon the growing notion that INs change their activity in relation to learning[5,14,16,41,42,44–46] in two ways: first, they indicate that SOMIs signal mixed information on internal engaged states and external task properties, and second, they show that SOMIs strongly support flex-ible behavioral changes in relation to novel environmental demands in animals engaged in the task. Age and anxiety can be excluded as additional factors that might have influenced behavioral performance (see section "Methods"). However, future work will be required to shed light on the potential relevance of social hierarchy and gender.

How can the dichotomous SOMI activity in relation to learning performance be reconciled? One of the common features of the DG afferent system is that it originates from a small number of neurons that exert a powerful control on the activity of a sparse group of GABAergic cells[47], which in turn have the ability to control the activity of large GC populations[48,49], representing contexts, cues and events[14,46,50]. The MSDB, with its role in controlling locomotion and experience-dependent context learning[51,52], innervates FSIs, pre-dominantly PVIs, as well as SOMIs in the DG[35,53,54]. Hilar SOMIs are the sole synaptic output from the DG to the MSDB[21]. Hence, reward-related information from the DG will inhibit PVIs, glutamatergic, and choli-nergic cells in the MSDB[21]. As SOMI-mediated inhibition gradually rises in anticipation of rewards in experts, it will be maximal in the MSDB close to the time of reward delivery. Locomotion can be reliably stopped in mice by pharmacological inactivation of the MSDB[55–57].

SOMI$_{proj}$-mediated inhibition at goal locations might therefore act as a 'velocity-down' signal in the MSDB. Consequently, mice will reduce their running speed and reach a minimum at the expected reward onset. Reduced feedforward excitation from the MSDB might, in turn, limit discharges of DG SOMIs as observed in CA1[39]. Thus, synaptic interactions between DG SOMIs and the MSDB are likely to define time windows of DG SOMI signaling and running velocity in accordance with the animals' attentive state. As glutamatergic inputs from the MSDB contact PCs and SOMIs in CA1-3[58–63], we propose that this regulatory mechanism may operate within the entire hippocampus. Noticeably, the fraction of active SOMI$_{proj}$ was markedly higher in experts, suggesting that the transmitted "velocity-down" signal will be stronger than in NEs, characterized by a lack of deceleration in pre-reward areas (Fig. 2d). As DG SOMIs expand some of their dendrites into the molecular layer[64], we expect that they receive goal-related multisensory inputs directly from the entorhinal cortex. This assumption fits SOMIs' high context-selectivity[14]. Recently, a novel selective GABAergic innervation of hippocampal SOMIs by the *pontine nucleus incertus* has been identified, which is markedly involved in contextual memory recall[65,66]. Thus, both cortical and subcortical inputs may refine SOMIs' activity and the encoding of anticipated goal-areas.

It is tempting to speculate that performance-related enhancement of DG SOMI activity might be tuned by the dynamics of reinforcement signals provided by neuromodulatory pathways. Subcortical neuromodulatory centers such as the dopaminergic *ventral tegmental area* (VTA), the noradrenergic *locus coeruleus*, and the serotonergic *raphe nuclei* have been implicated in signaling reward-related information to CA1[6,67–70], and to the neocortex[68,71,72]. Consistent with this proposal, the DG circuitry, particularly hilar INs, receives monosynaptic inputs from these brainstem regions[71,73–75]. Moreover, they can excite DG SOMIs[76,77], similar to observations in the neocortex[78]. All three brainstem areas are strongly interconnected with the *nucleus accumbens*, which plays a key role in the recall of spatial contexts during appetitive behavior[79]. Thus, future studies will provide the causal link between the dynamics of reinforcement signals and the activity dynamics of DG SOMIs during GOL-tasks in E compared to NE mice.

In addition to extra-hippocampal inputs, wiring of SOMIs within the local DG circuitry will also define their activity dynamics in relation to the animals' performance. GABAergic vasoactive intestinal polypeptide-expressing cells (VIPs) have been shown to target INs, particularly SOMIs in CA1[19,80–82], and the neocortex[83,84] and respond to reward-related reinforces[84]. The majority of CA1 VIPs reduce their activity upon reward onset in mice trained in a GOL task, although a minority displays elevated activity in response to reward delivery[42]. It is therefore likely that inhibitory and disinhibitory mechanisms provided by VIPs may contribute to SOMI signaling in the DG during GOL tasks. Moreover, SOMIs' loss in predictive goal coding upon reward removal at the originally trained goal locations could be supported by elevated activity of VIPs, which have been shown to detect novel experiences in CA1[82].

Is GC activity related to predictive SOMI signaling? GCs receive goal-related reinforcement signals from the entorhinal cortex[26]. Memorized reward locations restructure spatial representations in the entorhinal cortex by moving the fields of spatially- (grid) and object-tuned cells towards reward locations[85–88]. Such restructured input activity might shape the activity patterns of GC populations at reward locations. Although our study did not report GC and MC population activity at reward sites of E or NE mice, synaptic plasticity at GC-SOMI synapses might lead to synaptic strengthening[21,64], which in turn could amplify and maintain activity of GCs assembling at goal-sites, including place cells, through feedback dendritic inhibitory "winner-takes-all" mechanisms. Our finding that silencing of DG SOMIs results in the holding on to the "old memory" of previously rewarded locations is consistent with this interpretation (Fig. 5). Removal of dendritic

inhibition may allow the recall of "old memories" and the lacking ability to adjust behavior to novel contextual situations[14,89]. Additional future work will be required to enlighten the microcircuit mechanisms in the DG shaping the population activity dynamics, which in turn define the animals' behavior.

Taken together, our data demonstrate that DG SOMIs flexibly encode goal-locations in dependence of current and past experiences. In experts, SOMIs' activity dynamics predict the expected outcome, whereas in NEs their activity reflects reward consumption. These findings may have major implications in the role of the DG in memory and the control of reward-seeking behavior. The prevalent model proposes that outputs of dorsal CA1 to the brain stem are pivotal in inducing context-related reward-seeking behavior[70,79]. Our data expand on this model by demonstrating that DG SOMIs are important components of flexible goal-memory and related behavioral actions by encoding features of the task in high-performing animals and rapidly adapting the code according to task-changes[90].

## Methods

### Experimental methods

**Mice.** All animal procedures were performed in accordance with national and institutional guidelines and with the approval of the Regierungspräsidium Freiburg (License Nos: G19/035, G21/127). Adult male B6J.Cg-Ssttm 2.1(cre)Zjh/MwarJ and Ssttm2.1(cre)Zjh/J mice (SOM-Cre, 6–20 postnatal weeks, Jackson Laboratory, Stock Nos: 028864, 013044) were used. Groups of 2–4 mice were housed/cage on a non-reversed 12-h light-dark cycle at 20–24° and 45–65% humidity (procedures were conducted during light phases). Surgeries, recordings, and histology were performed using procedures adapted from previous publications of our laboratory[21,46,91]. Experiments were performed based on five cohorts of male mice in the same age range. Findings of Figs. 1–4 were performed in three independent replications with groups of 5-7 mice each. Data shown in Fig. 5 were repeated in two independent replications with four mice in controls and five mice for hM4Di-mediated SOMI silencing. In three out of five of the cohorts, E and NE mice were identified based on their foraging behavior. Mice were classified as experts if the following two criteria were fulfilled in the pre-reward area. First, the licking frequency was significantly enhanced 2 s prior to reward onset as compared to the baseline (2 s period in a time window of 4–6 s prior to pre-reward zone onset), and second, the running velocity was reduced in the same time window. Otherwise, mice were classified as NEs. We did not identify a relationship between age and performance levels in the task. Moreover, we did not observe in any of the tested animals anxiety-related behavior such as freezing or backwards running on the circular track. No statistical method was used to determine sample size prior to experiments. The experiments were not randomized, and the investigators were not blind to allocation during experiments and outcome assessment.

**Surgical procedures.** All surgical procedures were performed in a stereotactic apparatus (Kopf instruments) under general anesthesia induced by 3% and maintained by 1.5–2.5% isoflurane in $O_2$. Buprenorphine (0.1 mg/kg body weight) and carprofen (5 mg/kg body weight) were subcutaneously injected for analgesia before the surgery. To optogenetically identify DG SOMIs, a small bilateral craniotomy (0.5–1 mm diameter) was made over the hippocampus and AAV1-CAG-FLEX-hChR2(H134R)-mCherry (Charité Vector Core BA-049) or AAV1.CAGGS.Flex.ChR2.tdTomato. WPRES.SV40 (Addgene 18917) was injected bilaterally in the dorsal DG (anterior-posterior: −1.8 mm; medial-lateral: ±1.1 mm; dorsal-ventral: −2.2 mm). The craniotomies were covered by silicone sealant (Kwik-Cast, World Precision Instrument) to mark the coordinates for later surgeries on the recording days. In the same surgery session, a stainless-steel head plate was implanted on the skull and stabilized with dental cement (Super-Bond

C&B). A 200 μm core, 0.39 numerical aperture (NA; Thorlabs) optical fiber was implanted above the fimbria (anterior-posterior: −0.2 mm; medial-lateral: −0.26 mm; dorsal-ventral: −2.5 mm, 10° lateral angle), which was connected to a 488 nm elliptical dot laser (BioRay, Coherent) during recordings to activate ChR2. In addition, two screws were driven into the skull and covered with dental cement to strengthen the implant. Postoperative analgesic treatment was continued with buprenorphine (0.1 mg/kg body weight) and carprofen (5 mg/kg body weight) for 2 days after surgery. Mice were allowed to recover from surgery for 5–7 days before training sessions commenced.

After the animals met the behavioral criteria after training or were trained for maximal 20 days, a craniotomy (~2 mm) was performed under isoflurane anesthesia and carprofen analgesia (5 mg/kg body weight) above the dorsal DG of one hemisphere and sealed with Kwik-Cast sealant. The recordings started after at least 2 h of recovery in the home cage. Kwik-Cast was used to cover the craniotomy after each recording session. The same procedure was repeated on the other hemisphere on the following day. The expression of ChR2 and the position of the implanted fibers were verified post hoc following the end of the recording sessions.

**Virtual reality and behavioral training.** After post-surgery recovery, mice ($n = 27$) were housed in a big cage (35 × 45 × 21 cm) supplemented with a running disc, some nest-building materials, water ad libitum, and a limited amount of food to decrease their body weight by 10–15%. In the meantime, animals were handled daily and accommodated to the experimenter, recording setup and head fixation for 1–1.5 weeks before training. After habituation, mice were introduced to run on a circular Styrofoam wheel and exposed to a virtual reality (VR) provided by 5 screens covering ~210° of the animals' horizon and ~120° in elevation. A soymilk reward tube was placed close to the mouth of the animal. For our GOL task, animals were trained to move forward to navigate continuously in a virtual octangle maze (4 m length) and to seek rewards at 2 fixed reward locations on 2 opposite arms of the octagon. Reward zone (RZ) started 2 s prior and ended 2 s after soymilk delivery (8 μl, SMA Wysoy). The reward was delivered in the middle of the arm. Licking was monitored with an infrared beam breaker (EE-SX4070, Omicron Electronics) mounted on the reward tube. Rewards were delivered with a syringe pump (AL-1000, World Precision Instruments). The time point of reward delivery was defined as the time when a drop was delivered at the mouthpiece. This was empirically measured to occur 0.94 s after triggering the reward pump. We measured the sound spectrum and intensity of the pump during reward delivery with a sound meter (Precision Instruments, AL-1000) and detected a peak at ~11 kHz directly at the pump. However, this peak was not detected at the running wheel, suggesting that sound intensity declined over distance and could not be detected by the animal at the running wheel (distance between pump and wheel ~1.5 m). The pump sound is therefore unlikely to act as a conditioned cue that, paired with the reward, could drive SOMI activity. Recordings started after animals reliably consumed rewards or received training for 20 days. The habituation and training process lasted for 3–5 weeks. A subset of five mice were trained to navigate in an environment with random rewards for 3–5 days until they were transverse to the circular maze.

**Chemogenetic SOMI silencing.** Following our previously established protocol for the chemogenetic manipulation of SOMI activity[14], AAV2-hSyn-DIO- hM4D(Gi)-mCherry (titer 2.7*1012 vg ml-1: University of Pennsylvania Vector Core) was bilaterally injected into the dorsal DG of 4 SOM-Cre mice (Fig. 5). Clozapine control experiments were performed by injecting AAV2-hSyn-DIO-mCherry (titer 2.3 × 1012; University of Pennsylvania Vector Core) in 5 SOM-Cre mice. Animals were first trained for 15 days on the circular track in the virtual environment setup and thoroughly familiarized with the track. Lick rate and running velocity were recorded until mice learned to obtain rewards at defined

GOL locations and reached expert levels. After training, behavioral recordings (running velocity and lick rate) were started (20 min/day in the virtual reality). Clozapine (1 mg/kg; Tocris) was intraperitoneally injected. Animals were placed back in their home-cage for 15 min and then placed into the apparatus in which the reward site was translocated to novel areas (NOVs). The mice were recorded during 20 min/day for 2 subsequent days, corresponding to a set between 10 and 25 laps on the same track.

**In vivo electrophysiological recordings.** After post-surgery recovery on the recording day, mice were briefly anaesthetized with isoflurane and mounted on the running wheel. The silicon covering the surgical site was removed and the brain surface was perfused with saline. A 4-shank silicon probe consisting of 64 recording sites (16 sites/shank, 250 μm shank separation, P-1 probe; Cambridge Neurotech), attached to an optical fiber, mounted above the electrode sites and coated with fluorescent dye (DiD, Thermo Fisher Scientific), was inserted stereotaxically. After penetrating the brain surface, the silicon probe was first quickly advanced for ~1 mm (5–8 μm/s) and continued to be slowly moved (~1 μm/s) until reaching the hilus. Hippocampal cellular layers were identified physiologically by characteristic local field potential (LFP) activity patterns (e.g., sharp wave ripples for CA1 stratum pyramidal and dentate spikes (DSs) for DG and hilus). After reaching the final position of the probe, the probe and tissue settled for 10–15 min. Electrophysiological signals were sampled at 30 kHz with an Intan RHD2153 interface board and amplified with a 64-channel amplifier (Intan Technologies), connected to a universal serial bus acquisition board (OpenEphys). The location on the track was acquired in real time to control the time of reward delivery (Arduino Uno). All behavioral variables were simultaneously detected with the electrophysiological recordings (OpenEphys).

After 20–30 min of recordings in mice navigating in the familiar environment, an optogenetic stimulation protocol was administered by triggering an elliptical dot laser (488 nm; BioRay, Coherent) to identify SOMIs or other neurons influenced by SOMI-mediated inhibition. To differentiate local SOMIs (SOMI$_{local}$) from medial septal projecting ones (SOMI$_{proj}$), two optical fibers were implanted: one above the fimbria and the second one attached to the silicon probe above the DG. Optogenetic stimuli consisted of 50 ms or 250 ms rectangle pulses with 5 s inter-stimulus intervals and repeated ~100 times. Light intensity was 3–8 mW at the tips of the fibers and calibrated before fiber implantation or recording.

A subset of 11E animals were introduced to a reward translocation paradigm on the last day of recordings. The reward locations were translocated to the middle of the next arms of the octagon without changing the original virtual environment (wall patterns and objects remained the same). In a subset of experiments, we exposed mice to a novel VR (Supplementary Fig. 11). The novel context was characterized by different visual cues (wall patterns, floor, and objects) and lacking reward locations but the same dimensions as the FAM context (20 mice). To reveal recruitment of SOMIs by sensory stimuli, we applied discrete visual stimulation (flying object on the screen for 1 s), reward delivery, sound application (4 or 5 kHz for 1 s) and air-puff (500 ms) using a variable inter-stimulus interval between 3 and 20 s and the stimuli were repeated for 7–10 min in a pseudorandom order.

**Histology.** After the last recording session of the last animal in the holding group, mice were anaesthetized with overdosed intraperitoneal injection of ketamine/xylazine (ketamine: 120 mg/kg body weight, xylazine: 16 mg/kg body weight) and underwent trans-cardiac perfusion with phosphate-buffered saline (PBS) for 1 min, followed by 4% paraformaldehyde (PFA) for 12 min. The brain was dissected out and further fixed in 4% PFA overnight. Coronal or sagittal sections (100 μm thick) were cut from fixed brain samples using a vibratome (Leica), washed several times in PBS, and stained with DAPI. To verify

the specificity of viral infection in SOMIs, a subset of brain slices were incubated in anti-SOM antibodies (1:500, rabbit, Peninsula Laboratories LLC; LO A18PO21141; cat. No. T-4102; specificity was tested with a competitive ELISA) overnight at 4 °C, then washed several times in PBS before being transferred to secondary antibodies: rabbit-Cy3 (1:1000, rabbit, Jackson ImmunoResearch, UK, LOT 130329; cat. No. 111-165-003; specificity tested with immunoelectrophoresis and ELISA) for 2–2.5 h at room temperature. After incubation, the sections were washed, labeled with DAPI, and mounted on slides in the same orientation in the proper dorsal-ventral or medial-lateral order. The slices were scanned with a confocal laser-scanning microscope (LSM710 or 900, Zeiss). The quantification of viral-expressing specificity was determined based on the presence of SOM antibody signals and revealed its focus on the hilar DG and proximal CA3 without spread in distal CA3. Individual shank positions and the attached optical fiber were histologically identified based on DiD traces in the tissue and could be traced across the dorsal-ventral/medial-lateral order of sections and the tissue damage that was caused by the shanks or the fiber. The track of the implanted fiber over the fimbria was verified depending on the tissue damage.

## Quantitative analysis

**Spike extraction and sorting.** Spike sorting was performed with an automatic algorithm followed by manual curation, implementing autocorrelation and cross-correlation to obtain well-isolated units (Mountainsort [92]), as previously described[91]. Only clusters with a clear refractory period and clean waveform were kept for further analysis. Cross-correlation was applied for units presenting similar waveforms to assess whether they need to be merged. To extract the physical location of the units, the recording site where the largest amplitude of spikes was defined as the location of the unit and matched with the histology of the corresponding shank.

**Behavioral analysis.** Position data, recorded as a pulse-width modulated signal, was first low-pass filtered and, in some cases, manually curated to be transformed to a continuous signal. Speed data was obtained based on the position data. Periods with running speed <0.5 cm/s were considered as immobility while periods with velocity >3 cm/s as steady-state locomotion. The running onset was defined as epochs presenting a minimum of 3 s immobility (<0.5 cm/s) followed by a transition to reach velocity of 3 cm/s in ≤3 s. The opposite applied for the running offset. To determine rate changes of units in dependence of running speed, we aligned individual unit firing rate to velocity in 100 ms bins and performed linear least-squares regression. To further examine whether speed correlations were transient or persistent, we restricted to steady-state locomotion epochs and performed the same analysis as described above. To examine the robustness of the animals' reward consumption, we first detected the reward epochs and calculated the success rate as the percentage of trials in which the animal consumed rewards reliably. To further quantify the animals' learning performance, the lick rate of pre-, post-area (2 s before and after reward delivery, respectively) and the baseline, defined as 4–6 s before reward delivery, was calculated and the animal was considered as expert if consistent anticipatory licking behavior (significantly higher lick-rate in the pre-reward area compared to the remaining unrewarded track) was observed.

**Optogenetic tagging of Neurons.** To identify DG SOMIs and units modulated by SOMIs, blue light was delivered through the optical fiber attached to the silicon probe to the DG while recording single units and LFPs. Light-on epochs (50 and 250 ms; 70–100 repetitions respectively) were detected and spike numbers during these epochs were compared to those during the same interval prior to light delivery (pre-stimulation) for significance. Units were identified as SOMIs when spike numbers during light epochs were significantly higher than those

of pre-stimulation epochs, and the first light-triggered spikes occurred within 5 ms after light onset. A unit showing an increase within the 5 ms time window but a decrease after this time window during light-delivery was considered as a SOMI inhibited by other SOMIs. Cells showing significant increases during light epochs, but the first light-triggered spikes appeared with more than 5 ms delay after light onsets were considered as units disinhibited by SOMIs (SOM-disinh). Units presenting significant decreases during light epochs were considered to be inhibited by SOMIs (SOM-inh).

To further disambiguate SOMI$_{proj}$ from SOMI$_{local}$, the same stimulation protocol was applied to the fimbria, generating action potentials in long-range projecting axons of SOMI$_{proj}$. Identified SOMIs presenting significant increases by stimulating axons in the fimbria were defined as putative SOMI$_{proj}$, while those showing no significant changes were defined as putative SOMI$_{local}$. The same criterion was applied to units inhibited or disinhibited by SOMIs to uncover their specific recruitments by distinct SOMI subtypes.

**Unit classification based on physiological features and monosynaptic interactions of neuron pairs.** To reveal the identity of cells which are not putative SOMIs, we classified units recorded in the DG based on their physiological characteristics, assisted by histologically verified positions and monosynaptic excitatory and inhibitory interactions between simultaneously recorded units[32,93,94]. We first separated putative excitatory and inhibitory neurons on the basis of trough-to-peak duration and bursting index. To do so, we bandpass filtered raw data (0.3–6 kHz) and obtained the unit waveforms. Trough-to-peak duration was calculated by taking the average waveform of a given unit from the recording site with the highest amplitude. Bursting index was determined by taking the spike autocorrelogram and dividing the average spike number in the 3–5 ms bins by the average spike number in the 200–300 ms bins. Neurons with a high bursting index (>3) and large trough-to-peak duration (>0.45 ms) were characterized as putative excitatory units, whereas cells with a low bursting index (<3) and small trough-to-peak duration (<0.45 ms) were defined as putative fast-spike interneurons (FSI). Units presenting large trough-to-peak duration (>0.45 ms) but low bursting index (<3) were considered as wide-spike interneurons (WI). To further identify putative GCs and MCs, we applied k-mean clustering based on w-PC1, w-PC2 (obtained by PCA analysis on the second derivative of the sampled average waveform between 0 and 0.8 ms, 0 as the trough of the average spike waveform), the DS amplitude during phases of resting and spatial information (SI)[32,93]. The amplitude of DSs as well as its reversal at the GC layer-to-molecular layer boarder indicated the recording depth along the radial DG axis. Moreover, putative GCs displayed on average a significantly higher SI and lower mean firing rate than MCs and thereby confirmed previous GC and MC in vivo recordings[31].

Some PCs were identified based on putative synaptic connections using baseline corrected cross-correlation[95,96]. Two criteria were required to detect putative excitatory connections: (1) the peak of the cross-correlogram (CCG) of the spike trains exceeded that from the slowly co-modulated baseline, which was generated by convolving the observed CCG with a "partially hollow" Gaussian kernel (standard deviation: 10 ms; hollow fraction: 60% [95]); (2) the peak of the positive lags (causal direction) was significantly larger than the largest peak in the negative lags (anti-causal direction). A putative connection was considered significant when at least 2 consecutive bins in the CCG within +1.2 to +4 ms passed the statistical threshold by applying 99.9 percentile of the cumulative Poisson distribution. All detected monosynaptic interactions were manually inspected and refined after automatic detections.

**Analysis of behavioral event-related responses of single cells.** For GOL tasks and reward-translocation tasks, we quantified the activity of

individual cells in the reward zone, pre-, post-area, and the non-rewarded track (baseline). A cell presenting significantly higher or lower activity in the reward zone, pre- or post-area, compared to the baseline, was considered a reward-modulated unit. Individual cell activity was aligned by the start of the reward delivery in 100 ms bins with 3 s time intervals prior to and after the post-area and were smoothed by a Gaussian filter ($\sigma = 1$ bin) for visual display and the delay measurements. Smoothing has been applied only over time and not other parameters. To exclude the bias through units showing high firing rates, the firing rate of individual units was z-scored. Time delay was quantified as the time interval between reward onset and the maximal activity change in E and NE mice in the pre- or post-area. In Fig. 2f, this delay was measured for 37 out of 69 E cells and 8 out of 22 NE cells. We then calculated the absolute difference of the z-scored activity of the individual cells to the baseline 3 s prior to the reward onset. To reveal the influence of pure sensory stimuli on SOMIs, sensory stimulation epochs of single modality (reward, visual, sound and air-puff stimulation) were detected and spike numbers during these epochs were compared to those during the same interval prior to stimulation onsets for significance.

**General linear model predicting SOMI activity.** To test the contribution of behavioral parameters to SOMI activity near reward sites, we applied a general linear model to re-generate the SOMI activity as a linear function of the behavioral variables, including velocity, licking and reward. The spike trains of SOMIs of all trials were binned for the times corresponding to −3 to 3 s around the reward site to obtain a smoothed spiking signal (24 bins). Similarly, the binned lick rate velocity, and acceleration signals were obtained for the same time window. Reward was modeled as a binary signal, turning active at the reward delivery time point. Binned firing rates were predicted with the behavioral variables in a five-fold cross-validation regime using linear regression (LinearRegression of the scikit-learn package). The quality of the prediction was expressed as explained variance (EV). The predictions were compared against a surrogate distribution obtained by randomly permuting the behavioral variables (500 iterations). SOMIs for which the actual EV exceeded the 95th percentile of the surrogate distribution were considered to be significantly modulated. All further analysis was performed with the set of significantly modulated neurons. To assess the contribution of individual behavioral variables, a single variable was randomly permuted 500 times, and the obtained average EV was compared to the EV using the full model. The same procedure was also applied to the time window from 3 s before until reward onset to capture the anticipatory time window. In this model, reward was not considered.

**Decoder.** For single cell decoding (Supplementary Fig. 6), instantaneous firing rates of individual SOMIs (with >1 spike/1.5 s on average) were estimated in a baseline period (up to 1.5 s before reward delivery) and in an expectation period (from 1.5 s before reward delivery until 1.5 s after reward onset). For rate estimation, we used a kernel density estimator with a Gaussian kernel function of bandwidth 200 ms. Firing rate estimates were performed on half of the reward delivery trials (split into even and odd trials to account for non-stationarity). The other half of the trial was used for testing in a 2-fold cross-validation scheme. Test decoding accuracies (fraction correct) were obtained in a decoding window of variable length from 0.25 to 2.75 s such that for each spike count pattern of the test set, maximizing the corresponding time-resolved Poisson count log-likelihood led to a binary estimate of baseline vs. expectation period. If the fraction of correct estimates averaged over the 2 test folds (called accuracy) exceeded the 99-percentile of accuracies obtained from shuffled labels (baseline vs. expectation period), decoding was considered significant (white dots in Supplementary Fig. 6a indicate the smallest decoding window with significant decoding). Population decoding was performed by computing the sum of log-likelihoods over increasing subsets of cells independently in E and NE cohorts for varying decoding window lengths. Binary estimates of baseline vs. expectation period were based on maximum likelihood, using a 2-fold cross-validation scheme. To assess the effect of the behavioral variables (Supplementary Fig. 6c), we used the same binned activity as in the GLM and repeated the population decoding with modified binned spiking activity, where the modulation due to the speed and the acceleration signals, as predicted by GLM regression, was subtracted.

### Statistics
Data analysis (except initial spike sorting), statistical analysis, and figures were done with custom-made software in Python 2.7 and 3.7. All statistical tests were two-tailed. For comparisons between two populations, t-tests (unpaired t-test for unpaired samples; paired t-test for paired samples) were applied if the data points followed a normal distribution, assessed with a Shapiro–Wilk test. For data that were not normally distributed, the non-parametric Wilcoxon rank-sum test (for unpaired samples) or Wilcoxon sign-rank test (for paired samples) was performed. For multiple comparisons in the same population, data were analyzed using one-way analysis of variance (ANOVA; data normally distributed) or Friedman test (data not normally distributed) with repeated measures, followed by paired t-test (data with normal distribution) or Wilcoxon sign-rank test (data not normally distributed) with a Sidak or Bonferroni's correction whenever statistical significance was observed. Correlations were computed using Pearson's correlation coefficient or least-squares regression, noted in the text. Data are presented as mean ± SEM unless specified otherwise. Throughout the figures, P values are denoted by * ($P < 0.05$), ** ($P < 0.01$), and *** ($P < 0.001$). Boxplots represented median (red), mean (black dashed), and interquartile range (IQR, 25th–75th percentile), and whiskers extended to cover the distribution of data without outliers (defined as points above 1.5 IQR below or above the box edges). Bar-plots with lines and continuous lines with shadows represent mean ± SEM.

### Reporting summary
Further information on research design is available in the Nature Portfolio Reporting Summary linked to this article.

## Data availability
Relevant processed data sets have been deposited at https://gin.g-node.org/cleibold/somi_reward with the https://doi.org/10.12751/g-node.lh1rr2, to reproduce and verify the main results. All raw processed data sets for all figures are available through the corresponding Source data files. The raw data sets and any further information for the reanalysis of data reported in this paper will be made available from the lead contact (marlene.bartos@physiologie.uni-freiburg.de) upon request. There are no restrictions to the data availability. Source data are provided with this paper.

## Code availability
All custom-written codes have been deposited at GitHub (https://github.com/cleibold/somireward) [97] https://doi.org/10.5281/zenodo.15407729. Any additional information required to reanalyze the data reported in this paper is available upon request by the lead contact (marlene.bartos@physiologie.uni-freiburg.de).

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

## Acknowledgements

We thank Drs. T. Hainmueller, C. Elgueta, and T. Cholvin for reading earlier versions of the manuscript. We thank K. Winterhalter and K. Semmler for technical support, R. Verma for programming support for data analysis, and N. Vetter for animal training. This work was funded by the German Research Foundation (DFG BA1582/12-1 M.B.; FOR2143 M.B., CRC-TRR 384/1 2024, – 514483642 M.B., J.-F.S. and C.L.; FOR5159 M.B., J.-F.S. and C.L.) and by the ERC-AdG 787450 (M.B.).

## Author contributions

M.Y. and M.B. conceived the study; M.Y. performed and analyzed single-unit experiments. A.C. and M.B. designed the hM4Di experiments, A.C. performed hM4Di experiments, related data analysis, and behavioral testing. J.-F.S. performed the GLM analysis. S.G. and C.L. performed decoder analysis. M.Y. and M.B. wrote the manuscript. All authors edited the manuscript.

## Funding

## Competing interests

The authors declare no competing interests.
