## [Peer Review file · Nature Communications]

Predictive goal coding by dentate gyrus somatostatin-expressing interneurons in male mice

Corresponding Author: Dr Marlene Bartos

Version 0:

Reviewer comments:

Reviewer #1

(Remarks to the Author)

This study highlights the crucial role of somatostatin-expressing interneurons (SOMIs) in the hippocampal dentate gyrus in encoding reward-related spatial locations based on current and past experiences, ultimately shaping goal-directed behavior. Using advanced methodologies such as optotagging, single-unit recordings, and chemogenetic manipulations in mice performing a spatial goal-oriented learning task in virtual reality, the authors demonstrate that SOMIs' activity predicts reward locations. However, when reward locations are changed, SOMIs rapidly reconfigure their firing, shifting toward reward consumption. Whereas the study is supported by robust data obtained in well-designed experiments, emphasizing the flexibility of SOMIs in adapting behavior in dynamic environments, several questions and issues require to be addressed:

1. One interesting finding is that a significant proportion (~20%) of SOMI-INs were inhibited by optogenetic activation during optotagging. This suggests that SOMIs may exhibit strong interconnectivity, and the authors propose that SOMI projection neurons (SOMI_proj) receive inputs from local SOMI interneurons (SOMI_loc). However, another possibility is that this inhibition reflects spike inactivation upon excessive depolarization. Notably, Suppl. Fig. 2A shows that these neurons fire 1–2 action potentials at the start of light stimulation but are subsequently silenced. It is important for the authors to provide evidence that rules out spike inactivation as the cause of the observed inhibition. Doing so would strengthen the argument that the effect arises from SOMI-mediated inhibition through interneuron interactions.
2. Suppl. Fig. 2 requires more clarity on how the authors distinguish between SOMI direct activation and disinhibition-driven responses. Additionally, how is disinhibition of dentate gyrus units by SOMIs defined? Given that SOMIs are interconnected with other interneurons, such as fast-spiking interneurons (FSIs), the observed effects could potentially result from these broader network interactions. Addressing these points would refine the interpretation of the results.
3. Do the authors have data or references quantifying the proportion of SOMI_loc versus SOMI_proj cells in the DG? With approximately 50% of SOMIs targeted by viral vectors and 70% of these identified as projection neurons, there seems to be a preference for targeting SOMI_proj cells. Is there evidence of construct tropism that favors projecting neurons over local interneurons? Clarifying this would help in understanding whether the results reflect the natural distribution of these interneurons or an artifact of the viral approach.
4. The study reports differences in SOMI involvement between expert (E) and non-expert (non-E) mice. Several factors, such as age (given the wide range of 6–20 weeks), anxiety levels, attention level, or social hierarchy (established during group housing) might explain these differences in mouse learning. It would be useful for the authors to discuss these potential factors and propose why SOMIs are linked to anticipatory behavior (an executive function) in expert mice, while their activity shifts toward consummatory behavior in non-expert mice.
5. A key finding of the study is the rapid loss of SOMI predictive activity when the environment changes. This raises the question of what mechanisms underlie such rapid reconfiguration of the circuit. A deeper discussion of potential mechanisms would add valuable insight.

6. The loss of predictive goal-coding by SOMIs following goal translocation may reflect enhanced inhibition by VIP interneurons, which are known to encode novel experiences in the hippocampus, particularly in CA1 (Tamboli et al., 2024). Although similar mechanisms in the DG are yet to be confirmed, it would be valuable for the authors to explore this possibility as part of the discussion.

7. To reflect the hippocampal area specificity of the findings, I recommend revising the title to: "Predictive Goal Coding by Dentate Gyrus Somatostatin Interneurons".

Minor:

- Line 459: Correct "of a spars group" to "of a sparse group."

- Lines 497–499:

Rephrase: "The majority of CA1 VIP interneurons reduce their activity upon reward onset in mice trained in a goal-oriented learning (GOL) task, although a minority show elevated activity in response to reward delivery." According to Turi et al. (2019), VIP interneuron modulation by reward depends on their speed modulation: positively speed-modulated cells tend to be inhibited by reward, while negatively speed-modulated cells are typically excited. Additionally, the spatial memory-dependent GOL task enhances CA1 VIP interneuron activity specifically during running stops when a reward is present.

Reviewer #2

(Remarks to the Author)

This manuscript by Yuan et al., investigates the role of dentate gyrus somatostatin-expressing interneurons (SOMIs) in reward coding. How different hippocampal cell types contribute to spatial and reward coding is a central question in the field. While extensive research has been done in this topic, most of it has been focused on the role of principal excitatory cells, and much less on the role of specific inhibitory interneurons, especially those in the dentate gyrus. To address this question the authors conducted silicon probe recordings in the dentate gyrus, together with optogenetic identification of SOMIs, while head-fixed mice performed a virtual reality spatial learning task. The authors found that the activity of SOMIs ramped up right before animals reached previously learned reward locations, and rapidly reconfigured when those locations were changed. Silencing of SOMIs affected behavioral performance in novel goal learning. The manuscript is clear and well written, the logic of experiments and analysis is sound, and the results presented support the authors interpretation. These results will be of interests for the broader community. The points below detail several additional analysis and controls that are needed to clarify some points and strengthen the authors interpretations.

1. The authors provide an exciting dataset of optogenetically identified dentate gyrus SOMIs. They used a clever way of characterizing two different subtypes of DG SOMIs, SOMIsproj and SOMIslocal, going beyond the usual in this type of experiments. However, their present results do not show important functional differences among these cell types. It would be useful to take advantage of this dataset and perform a more in-depth characterization of physiological and functional properties of these two cell types. This analysis could include an examination of their functional coupling with other dentate gyrus cell types, such as granule and mossy cells, or with LFP patterns. Although a bit tangential to the main story it would be a useful resource for many readers.

2. The central and most interesting finding in the paper is the involvement of DG SOMIs in predictive goal coding and flexible goal behavior. Several additional analyses are needed to clarify the interpretation of the findings and rule out potential alternatives.

a. The ramping up in SOMI activity before reward delivery is interesting, which leads to the interesting proposal that SOMIs provide "predictive encoding of expected outcomes". However, the animals also exhibited anticipatory behaviors (such as anticipatory licking and a drop in speed) coincident with the changes of SOMI activity (Figs. 2 and 4). While the authors performed several controls to examine the contribution of these behavioral variables, in its present form they cannot be fully evaluated. Using a GLM is a good way to address this issue. The authors focused their GLM analysis in comparing expert with non-expert mice. While an interesting comparison, they should also independently examine the contribution of behavioral variables and reward to SOMIs predictive coding in expert mice. They should explicitly test whether there is still significant explained variance after removing speed and lick rate variables (a statistical test currently missing in Fig S5). Also, to specifically test their impact in predictive coding, it would be better to use the temporal window preceding the reward (e.g., -3 to 0 s) rather than a longer window (-3 to 3s).

b. A large fraction of SOMIs exhibited speed-modulated activity (Fig. 3), consistent with previous reports that speed is a strong modulator of hippocampal activity. To control for the speed confound, the authors include an analysis showing that the activity of SOMI cells did not change dramatically during the steady running stage (Fig. S8). However, the running speed itself was also stable during this stage, providing a very narrow dynamic range for measuring speed modulation. To provide more further controls, the authors can include speed into their estimation model for the SOMI firing rate (e.g., Chiossi et al. 2024; using GLM or multiple regression), and exclude the speed contribution to firing rate before comparing E vs. NE SOMI activity. It would be also useful to test whether acceleration is a stronger modulator of SOMI firing than speed per se.

c. To better understand the specific contribution of SOMIs to reward coding the authors could show in a similar way the firing of the other cell types present in their recordings.

Antonio Fernandez-Ruiz

Reviewer #3

(Remarks to the Author)

In this manuscript Yuan and colleagues record from somatostatin-expressing inhibitory neurons (SOMIs) in the dentate gyrus of mice as they run on a circular maze for rewards. Using silicon probes they record neural activity and optogenetically identify SOMIs. They find that in well-trained animals just over half of the SOMIs are activated prior to reward, suggesting they may be involved in reward anticipation. When the reward is shifted, the anticipatory licking at the previously rewarded location is reduced. In animals that did not show anticipatory licking, these responses were absent and SOMIs were active only after reward consumption. Finally, the authors inhibited SOMIs and found reduced flexibility in licking responses, suggesting that SOMIs may play a causal role in mediating flexible spatial behaviors.

Overall the manuscript is interesting, with a clear rationale for examining SOMIs during spatial exploration. There is substantial novelty in recording DG SOMIs with electrophysiology, characterizing them as local or projection, and directly inhibiting them. While there is some conceptual overlap with recent work (including from this group), the manuscript is still of broad interest as the specific role of SOMIs, particularly in DG, is still poorly understood. The experiments are generally well reasoned, technically sound, and rigorous. However, several weaknesses limit the interpretability of the data, but could be addressed with additional data.

Major Comments:

1. The results from the chemogenetic inhibition experiment, as presented, do not clearly support the major premise of the manuscript, that DG SOMIs support flexible spatial behavior. The key metrics that are used to evaluate learning in this task (e.g., comparison of lick rates in the pre-reward zone to a baseline period, lick rates at the novel location) are not presented, and there appear to be changes in overall lick rates that make it difficult to interpret the results. There also appears to still be pre-emptive licking in the hM4Di group, which undermines the stated conclusions.

Additional Comments:

2. Line 172 - "suggesting that licking per se had no apparent effect on SOMI firing." It still seems unclear whether licking is contributing to the SOMI response. While the licking rate outlasts the SOMI activation, the timing of lick initiation and SOMI firing is highly correlated and this certainly does not rule out that the two are related. In general, the use of anticipatory licking as a measure of behavior limits the experimental manipulations that could dissociate licking from performance.

3. A characterization of the viral expression and spread is necessary to interpret the hM4Di experiment

4. Many of the plots are heavily smoothed, and it appears they may even be smoothed across dimensions that should not be (e.g., across animals in Fig 2B or cells in Fig 3). Please clarify whether all smoothing is done only over time/space.

5. The authors focus on local and projection SOMIs but there are certainly more than 2 subtypes of SOMIs in DG hilus, which is clear from the variable responses observed in Fig 2E. Additional discussion of these, or perhaps a characterization of the firing properties of reward anticipatory versus non-responding SOMIs could be helpful.

6. Since the reward pump is triggered ~1 sec before the reward is given (line 568), is it possible the SOMIs are responding to a conditioned auditory cue from the pump? The random sound cues presented in Supplemental Fig 6B partially addresses this, but it is still possible that a conditioned cue previously paired with reward could drive SOMI activity. (also see next comment)

7. Supplemental 6B (Sound) - how can this example cell have a baseline firing rate around 150hz? It is dramatically different than all other cells. This does not seem physiological and undermines the conclusion of this panel. Please clarify this.

8. Additional details on how Experts and Non-Experts were split are needed in the results and methods sections.

9. Line 304 - "To further proof that SOMI firing rapidly changes in ORI-zones during relocation" - This wording needs to be changed.

10. Fig 5 - The viruses appear to be labeled incorrectly in this figure and caption. Please clarify if Control virus was also a FLEX virus.

11. Line 351 - "If DG SOMIs provide a confirmation signal of the expected outcome..." This was very confusing as this idea was not really introduced properly. Why do the authors hypothesize it is a confirmation signal?

12. Fig 5D - what are the different colors within each subpanel? Please clarify

13. Could this "reward prediction" be driven by feedback input from overrepresentation of place cells near rewards? Is it possible to decouple place cell overrepresentation and this predictive coding? A discussion of this would be helpful.

Reviewer #4

(Remarks to the Author)

Version 1:

Reviewer comments:

Reviewer #1

(Remarks to the Author)

The authors have thoroughly addressed the critical comments and made significant revisions to clarify key sections and enhance the interpretation of the data. The manuscript has improved considerably as a result. I have no further suggestions.

(Remarks on code availability)

Reviewer #2

(Remarks to the Author)

The authors have addressed all the points I raised in my original revision. Crucially, they now provide extensive new additional statistical tests and controls that strongly support the specific modulation of SOMIs activity by reward expectation. They also show that this modulation is not present in non-expert mice and in other cell types such as fast-spiking interneurons. I have no further issues to raise and recommend this paper for publication. I believe it will be an important contribution to the field.

A small point regarding the characterization of SOMIs subtypes. The authors found no differences in some of their physiological properties (like modulation by LFP oscillations) and decided not to include them in the manuscript. I still believe this is useful information for future readers and worth including, but I leave it to the authors discretion on whether they prefer not to do it.

(Remarks on code availability)

Reviewer #3

(Remarks to the Author)

The authors have done an excellent job of addressing my comments and those of the other reviewers. I think this is an extremely valuable contribution to our understanding of inhibitory control of flexible behaviors. I have only 1 minor comment: The revised manuscript includes rates of connectivity between mossy cells, granule cells, and SOMIs. The methods to differentiate these cell types was included in the author rebuttal, but should be included in the methods of the manuscript as well.

(Remarks on code availability)

Reviewer #4

(Remarks to the Author)

(Remarks on code availability)

All changes in the manuscript are highlighted in yellow.

Reviewer #1 (Remarks to the Author):

This study highlights the crucial role of somatostatin-expressing interneurons (SOMIs) in the hippocampal dentate gyrus in encoding reward-related spatial locations based on current and past experiences, ultimately shaping goal-directed behavior. Using advanced methodologies such as optotagging, single-unit recordings, and chemogenetic manipulations in mice performing a spatial goal-oriented learning task in virtual reality, the authors demonstrate that SOMIs' activity predicts reward locations. However, when reward locations are changed, SOMIs rapidly reconfigure their firing, shifting toward reward consumption. Whereas the study is supported by robust data obtained in well-designed experiments, emphasizing the flexibility of SOMIs in adapting behavior in dynamic environments, several questions and issues require to be addressed:

We would like to thank the reviewer for her/his positive assessment of our work and praising the usage of advanced methods and the robustness of our data.

1. One interesting finding is that a significant proportion (~20%) of SOMI-INs were inhibited by optogenetic activation during optotagging. This suggests that SOMIs may exhibit strong interconnectivity, and the authors propose that SOMI projection neurons (SOMI_proj) receive inputs from local SOMI interneurons (SOMI_loc). However, another possibility is that this inhibition reflects spike inactivation upon excessive depolarization. Notably, Suppl. Fig. 2A shows that these neurons fire 1–2 action potentials at the start of light stimulation but are subsequently silenced. It is important for the authors to provide evidence that rules out spike inactivation as the cause of the observed inhibition. Doing so would strengthen the argument that the effect arises from SOMI-mediated inhibition through interneuron interactions.

We thank the reviewer for her/his insightful comments. In our previous study (Yuan et al., eLife 2017), we showed that large amplitude current injections for 1 second during *in vitro* whole-cell recordings resulted in maximal discharge frequencies of DG SOMIs at ~140 Hz with mild spike adaptation (~2.1, last inter-spike-interval divided by the first inter-spike-interval) and only a mild reduction in the peak amplitude of individual subsequent action potentials in a train. Moreover, whole-cell *in vitro* recordings from SOMIs expressing channelrhodopsin-2 during blue light application did not show a sudden loss of discharges (**Figure 1** for reviewers). Thus, we strongly believe that inactivation of action potentials is a very unlikely scenario for the observed results. However, as we cannot fully exclude the possibility of inactivation of voltage-gated ion channels in SOMIs *in vivo*, we toned down our statement on the neuronal silencing of SOMIs by presynaptic SOMIs on **line 90** of the results section of the revised manuscript and in the Figure legend of Supplementary Figure 2, **line 37-44** of the revised extended data.

Figure 1 for reviewers: Whole-cell patch clamp recording of a DG SOMI expressing channelrhodopsin-2 in acute hippocampal slice preparations of SOM-Cre mice show lacking spike accommodation and adaptation during blue light application. (A) Blue light application for 500 ms evoked a train of action potentials. A short light pulse of 2 ms evoked individual action potentials (7 spikes are superimposed). (B) *Left*, light pulses at 50 Hz for 2 ms to the slice evoked inhibitory postsynaptic currents (IPSCs) in the recorded granule cell (GC). The GC was identified based on the accommodating spike train evoked by positive current injection (upper trace) and its soma location in the granule cell layer. *Right*, same as left but the recording was performed from a fast-spiking DG interneuron. (Yuan & Bartos, unpublished observations).

2. Suppl. Fig. 2 requires more clarity on how the authors distinguish between SOMI direct activation and disinhibition-driven responses. Additionally, how is disinhibition of dentate gyrus units by SOMIs

defined? Given that SOMIs are interconnected with other interneurons, such as fast-spiking interneurons (FSIs), the observed effects could potentially result from these broader network interactions. Addressing these points would refine the interpretation of the results.

We thank the reviewer for pointing to this lacking information. SOMI activation is defined by the instantaneous and trial-by-trial reliable recruitment of action potentials upon light onset whereas disinhibition is characterized by a delayed and less reliable recruitment with a delay of > 5 ms after light onset. Considering a synaptic latency between presynaptic action potential generation and postsynaptic signaling of ~5 ms, disynaptic disinhibition should occur after a delay of at least 5 ms from light onset. We further expect that disynaptic disinhibition might be temporally less precise than direct light-mediated activation. We mention potential network effects causing disinhibition of units either by inhibition of presynaptic SOMIs or other types of GABAergic cells such as fast-spiking parvalbumin-expressing interneurons (PVLs) by light-activated SOMIs and refer to the paper of Savanthrapadian et al., J Neurosci, 2014, which shows that both interneuron types are interconnected **on page 3, line 91-94**, and the legend of **Supplementary Figure 2 (lines 38-45)** of the revised manuscript.

3. Do the authors have data or references quantifying the proportion of SOMI_loc versus SOMI_proj cells in the DG? With approximately 50% of SOMIs targeted by viral vectors and 70% of these identified as projection neurons, there seems to be a preference for targeting SOMI_proj cells. Is there evidence of construct tropism that favors projecting neurons over local interneurons? Clarifying this would help in understanding whether the results reflect the natural distribution of these interneurons or an artifact of the viral approach.

We found in our previous study that 8 out of 39 (20.5%) SOMIs were characterized by axon arborizations in the outer molecular layer and 24 out of 39 (62%) showed long-range projections to the medial septum (Yuan et al., eLife 2017). These data match our here made quantification of 30% SOMI local vs 70% SOMIs long-range. This information is now added on **lines 99-100** of the revised manuscript. Construct tropism is therefore an unlikely cause underlying our findings.

4. The study reports differences in SOMI involvement between expert (E) and non-expert (non-E) mice. Several factors, such as age (given the wide range of 6–20 weeks), anxiety levels, attention level, or social hierarchy (established during group housing) might explain these differences in mouse learning. It would be useful for the authors to discuss these potential factors and propose why SOMIs are linked to anticipatory behavior (an executive function) in expert mice, while their activity shifts toward consummatory behavior in non-expert mice.

All single unit recordings have been performed in male mice. Five cohorts of mice were tested and each cohort had animals of the same age range of 6-20 postnatal weeks. Although animals belonging to the same cohort were trained and housed under exactly the same conditions, we still found in 3 out of 5 cohorts both expert and non-expert mice (in the remaining 2 only experts). We did not observe a relationship between age and expert level. Animals have been habituated and trained for 3-5 weeks in the virtual reality and they did not show anxiety-related behavior such as freezing or backwards running. We therefore exclude anxiety or age as potential factors underlying different learning behavior. This is now explicitly stated in the Methods section on **page 17, lines 572-579**. We did not test for social hierarchy or gender-related differences and discuss these potential factors on **page 15, lines 493-495** of the revised manuscript.

5. A key finding of the study is the rapid loss of SOMI predictive activity when the environment changes. This raises the question of what mechanisms underlie such rapid reconfiguration of the circuit. A deeper discussion of potential mechanisms would add valuable insight.

Following the recommendation of the reviewer, we extended our discussion on **page 16, lines 533-537** and highlight the potential role of vasoactive intestinal polypeptide-expressing (VIP) cells, which inhibit

SOMIs in CA1 and are recruited by novel experiences and, thus, may silence DG SOMIs. In this context, we refer to Tamboli et al., Cell Rep, 2024. We would also like to refer to our discussion on potential interactions with local GCs on line 546-553 as well as the influence by neuromodulators on line 528-538.

6. *The loss of predictive goal-coding by SOMIs following goal translocation may reflect enhanced inhibition by VIP interneurons, which are known to encode novel experiences in the hippocampus, particularly in CA1 (Tamboli et al., 2024). Although similar mechanisms in the DG are yet to be confirmed, it would be valuable for the authors to explore this possibility as part of the discussion.*

We thank the reviewer for this proposal, which we took up on **page 16, pages 537-540**. In this context, we cite Tamboli et al. Cell Rep, 2024. See also our response above to point 5.

7. *To reflect the hippocampal area specificity of the findings, I recommend revising the title to: "Predictive Goal Coding by Dentate Gyrus Somatostatin Interneurons".*

We thank the reviewer for her/his advice and altered the title accordingly.

Minor:

- *Line 459: Correct "of a spars group" to "of a sparse group."*

Done. Now **line 498**.

- *Lines 497–499:*

Rephrase: "The majority of CA1 VIP interneurons reduce their activity upon reward onset in mice trained in a goal-oriented learning (GOL) task, although a minority show elevated activity in response to reward delivery." According to Turi et al. (2019), VIP interneuron modulation by reward depends on their speed modulation: positively speed-modulated cells tend to be inhibited by reward, while negatively speed-modulated cells are typically excited. Additionally, the spatial memory-dependent GOL task enhances CA1 VIP interneuron activity specifically during running stops when a reward is present.

We thank the reviewer for the clarifying comments. We rephrased the sentence accordingly.

Reviewer #2 (Remarks to the Author):

This manuscript by Yuan et al., investigates the role of dentate gyrus somatostatin-expressing interneurons (SOMIs) in reward coding. How different hippocampal cell types contribute to spatial and reward coding is a central question in the field. While extensive research has been done in this topic, most of it has been focused on the role of principal excitatory cells, and much less on the role of specific inhibitory interneurons, especially those in the dentate gyrus. To address this question the authors conducted silicon probe recordings in the dentate gyrus, together with optogenetic identification of SOMIs, while head-fixed mice performed a virtual reality spatial learning task. The authors found that the activity of SOMIs ramped up right before animals reached previously learned reward locations, and rapidly reconfigured when those locations were changed. Silencing of SOMIs affected behavioral performance in novel goal learning. The manuscript is clear and well written, the logic of experiments and analysis is sound, and the results presented support the authors interpretation. These results will be of interests for the broader community. The points below detail several additional analysis and controls that are needed to clarify some points and strengthen the authors interpretations.

We thank the reviewer for praising the quality of our data and the interest of our work to a broader neuroscience community.

1. The authors provide an exciting dataset of optogenetically identified dentate gyrus SOMIs. They used a clever way of characterizing two different subtypes of DG SOMIs, SOMI_{sproj} and SOMI_{slocal}, going beyond the usual in this type of experiments. However, their present results do not show important functional differences among these cell types. It would be useful to take advantage of this dataset and perform a more in-depth characterization of physiological and functional properties of these two cell types. This analysis could include an examination of their functional coupling with other dentate gyrus cell types, such as granule and mossy cells, or with LFP patterns. Although a bit tangential to the main story it would be a useful resource for many readers.

In our previous *in vitro* work (Yuan et al., eLife 2017) we demonstrated that the two SOMI types display different morphological, intrinsic passive and active membrane properties. This work further showed that SOMI_{local} are connected to GC dendrites while SOMI_{long} to somata of SOMIs and fast-spiking parvalbumin-expressing interneurons (PVIs). Based on spike correlation analysis we revealed only 4 putative GC-SOMI and 3 MC-SOMI connections with GCs/MCs that showed elevated activity at reward sites (**Figure 2** for reviewer). This number is too low for a rigorous quantitative analysis. We therefore decided not to include it in the current manuscript. However, we found that in the total of 21 putative PC-SOMI pairs, GCs preferred to target SOMI_{local} (4 out of 5 pairs) and MCs preferentially targeted SOMI_{proj} (7 out of 7 pairs). In the remaining 9 Pc-SOMI pairs the optogenetic differentiation between local vs long-range projecting SOMIs was not clear. This information is now included on **page 4, lines 113-114** of the revised manuscript. In Supplementary Figure 3, we plotted the bursting index as a function of the

trough-to-peak duration of single units from SOMI_{local} and SOMI_{long} (orange vs green circles) and observed no difference in the single unit kinetic properties. We also started to examine the relationship of single units to oscillatory cycles during theta (4-12 Hz) and gamma (30-100 Hz) network activity patterns and observed no marked differences in the preferred average angle of SOMI subtypes to the respective theta and gamma cycles (**Figure 3** for reviewers). Based on the lacking differences, we decided not to present the figure in the revised manuscript and hope that the reviewer can agree on this decision.

Figure 2 for reviewers: Single unit correlation analysis for putative GC-SOMI and MC-SOMI pairs. DG SOMIs receive glutamatergic inputs from GCs and mossy cells (MCs; Yuan et al., 2017; eLife). Here, we used cross-correlation analysis to detect mono-synaptically connected units. The latency between putative presynaptic GCs and opto-tagged SOMIs was 2.5 ± 0.15 ms (4 GC-SOMI and 3 MC-SOMI pairs).

Figure 3 for reviewers: Lacking difference in the preferred theta and gamma angles of SOMI_{loc} and SOMI_{proj}. (A) Timing of an optogenetically identified SOMI unit in relation to the local field potential (LFP) in the hilar DG filtered with different band-pass filters. (B) Circular plots depict SOMI subtype clusters during theta (*upper left*), low gamma (30-50 Hz, *upper right*), mid gamma (50-70 Hz, *lower left*) and high gamma (70-100 Hz, *lower right*) network oscillations and show their preferred angles to the corresponding oscillatory cycles. Colored lines relate to the mean cluster angles defined by the SOMI subtypes. 'All SOMIs' refers to cells, which could not be unequivocally identified as local vs projecting SOMIs. There is a lacking significant difference in the preferred average angle between SOMI_{loc} and SOMI_{proj} (Kuiper tests for circular distributed variables taking into account only significantly modulated cells in the respective frequency band: theta $P = 1.0$, SOMI-local = 24, SOMI-proj = 64; low gamma $P = 1.0$, SOMI-local = 14, SOMI-proj = 35; mid gamma $P = 1.0$, SOMI-local = 14, SOMI-proj = 27; high gamma $P = 1.0$, SOMI-local = 15, SOMI-proj = 48).

2. *The central and most interesting finding in the paper is the involvement of DG SOMIs in predictive goal coding and flexible goal behavior. Several additional analyses are needed to clarify the interpretation of the findings and rule out potential alternatives.*

a. *The ramping up in SOMI activity before reward delivery is interesting, which leads to the interesting proposal that SOMIs provide “predictive encoding of expected outcomes”. However, the animals also exhibited anticipatory behaviors (such as anticipatory licking and a drop in speed) coincident with the changes of SOMI activity (Figs. 2 and 4). While the authors performed several controls to examine the contribution of these behavioral variables, in its present form they cannot be fully evaluated. Using a GLM is a good way to address this issue. The authors focused their GLM analysis in comparing expert with non-expert mice. While an interesting comparison, they should also independently examine the contribution of behavioral variables and reward to SOMIs predictive coding in expert mice. They should explicitly test whether there is still significant explained variance after removing speed and lick rate variables (a statistical test currently missing in Fig S5). Also, to specifically test their impact in predictive coding, it would be better to use the temporal window preceding the reward (e.g., -3 to 0 s) rather than a longer window (-3 to 3s).*

Thank you for this excellent proposal, which we followed. To further disentangle the potential contribution of the key behavioral variables licking, speed and reward delivery to SOMIs' activity, we first applied a general linear model (GLM). The model revealed a larger proportion of SOMIs for which the activity could be significantly explained by the behavioral parameters in E as compared to NE animals (Fig. 2I). To further dissect the impact of the different variables in E mice, we followed the reviewers advice and constructed reduced models in which one of the variables was randomly permuted. This analysis revealed a significant effect of licking and reward but not of speed or acceleration. These data are now shown in the revised **Supplementary Figure 5c**. To further assess whether SOMI activity can be explained during anticipation of rewards, we ran the same analysis for a time window from 3 s before the reward until reward onset. Similarly, only licking but not speed or acceleration significantly affected SOMI activity in E mice during the anticipatory window. These data are shown in the revised **Supplementary Figure 5d**. Thus, our data show that SOMI activity in experts at goal-locations is modulated by anticipatory behaviour.

To directly address whether SOMI activity patterns bear information on anticipated reward locations, we applied the inverse approach and used differences between baseline activity (time before pre-reward area) and the activity recorded in intervals of varying lengths starting 1.5 s prior to reward onset to predict rewards, by using a cross-validated maximum likelihood (ML) decoder. As shown in the new **Supplementary Figure 6**, significant reward prediction was achieved on the single cell level by 30% of the recorded cells (>1 spike/1.5 s on average) in experts before reward onset but for none of the SOMIs in non-expert mice because no significant rate changes were observed during the pre-reward period (**Supplementary Figure 6a**). Extending the ML decoder to the population level as shown in the new **Supplementary Figure 6b**, we found that only 10 SOMIs were necessary to predict rewards with 80% accuracy in experts based on a 1.5 s period before reward onset. This criterion was not reached in non-experts even if we included all recorded SOMIs (**Supplementary Figure 6b**, lower panel). Repeating

the same analysis with activity traces from which speed and acceleration effects were subtracted according to GLM fits revealed successful decoding at 75% prediction performance (**Supplementary Fig. 6c**).

- b. *A large fraction of SOMIs exhibited speed-modulated activity (Fig. 3), consistent with previous reports that speed is a strong modulator of hippocampal activity. To control for the speed confound, the authors include an analysis showing that the activity of SOMI cells did not change dramatically during the steady running stage (Fig. S8). However, the running speed itself was also stable during this stage, providing a very narrow dynamic range for measuring speed modulation. To provide more further controls, the authors can include speed into their estimation model for the SOMI firing rate (e.g., Chiossi et al. 2024; using GLM or multiple regression), and exclude the speed contribution to firing rate before comparing E vs. NE SOMI activity. It would be also useful to test whether acceleration is a stronger modulator of SOMI firing than speed per se.*

We thank the reviewer for this proposal, which we followed. We include additional GLM analysis by excluding speed and acceleration from the model and compared expert and non-expert mice. Licking explained SOMI activity in E but not in NE mice but neither running speed nor acceleration (please see revised **Supplementary Figure 5d** and our above response to sub-point a). We also provide newly analyzed SOMI data in the **new Supplementary Figure 10** of the revised manuscript demonstrating that the majority of SOMIs is not modulated by acceleration.

- c. *To better understand the specific contribution of SOMIs to reward coding the authors could show in a similar way the firing of the other cell types present in their recordings.*

Following the reviewers' advice, we now show in the **new Figures 4 for reviewers** the activity dynamics of putative GCs and MCs at goal sides. First, we identified putative GCs and MCs by applying k-mean clustering based on w-PC1, w-PC2 (obtained by PCA analysis on the second derivative of the sampled average waveform between 0 and 0.8 ms, 0 as the trough of the average spike waveform), the dentate spike (DS) amplitude and spatial information (SI) following Senzai & Buszáki, Neuron 2017 and Kim et al., Nature Com 2020. The amplitude of dentate spikes as well as its reversal at the granule cell layer-to-molecular layer boarder indicated the recording depth along the radial DG axis. Second, we found that putative GCs displayed on average a significantly higher SI and lower mean firing rate than MCs and thereby confirmed previous GC and MC *in vivo* recordings (see panel C; Goodsmith et al., Neuron 2017; Danielson et al., Neuron 2017). Third, we observed that similar to SOMIs, the proportion of active GCs and MCs that accumulated at reward areas was substantially larger in experts as compared to non-experts (panel C,D,F). Moreover, GCs and MCs showed heterogeneous activity dynamics at goal locations in E mice with some of the cells maximally increasing and others declining their firing rates after reward onset (green vs orange lines, respectively, in panel C,D). Finally and similar to SOMIs, reward translocation (TR) from the FAM to new previously unrewarded areas on the same track, resulted in the absence of active GCs and MCs at the new TR zone as well as their absent activity at previously rewarded original area (ORI, **Figure 5 for reviewers**). Thus, our data show that GC and MC populations encode goal-sides in experts and rapidly reconfigure their activity if the expected outcome has not been confirmed. As these data raise several questions, (e.g. who inhibits GCs/MCs at FAM sites? Why does the peak of active GCs emerge earlier than the activity minimum of silenced GCs?), we decided not to present them in the revised manuscript but to increase the number of recordings, improve our analysis and try to explain the activity changes by GCs and MCs at reward locations. We would prefer to include the data in a follower manuscript.

We further analyzed the activity of fast-spiking (FS) interneurons in our dataset, which we have included in a **new Supplementary Figure 13**. This analysis shows that in marked contrast to SOMIs, the activity of FS interneurons does not predict future target locations. As the majority of FS interneurons are positively speed-modulated, they reduced their mean discharges in experts and in non-experts as mice approach the reward locations (**Supplementary Figure 13a-c**). Indeed, the time course of activity changes of FS-interneurons closely reflected the animals' running speed at the FAM and NOV reward

zones, which is consistent with their primary positive speed modulation in the DG (PVIs; Hainmueller et al., 2024, Nature Comm 15). Therefore, FS-interneurons reached their minimum activity slightly earlier in expert than in non-expert animals. Thus, predictive goal coding involves SOMIs but not FS-interneurons. This information is now included on **page 12, lines 379-383 & line 446**.

Figure 5 for reviewers: Reward translocation affects reward expectation-related GC and MC activity in expert mice. (A) Top, schematic representation of reward-translocation experiments. *Below*, green/orange line with shadow represents the mean \pm SEM activity in relation to reward onset at time 0 (vertical green striped line) of putative GCs in the familiar (FAM) reward area, after translocation to a novel previously unrewarded zone (TR) and in the original FAM zone after reward translocation (ORI). **(B)** Same as A, but for putative mossy cells (MCs). 3 mice, 11 GCs, 5 MCs. Braun square depicts the pre-reward area, gray square the post-reward zone.

Reviewer #3 (Remarks to the Author):

In this manuscript Yuan and colleagues record from somatostatin-expressing inhibitory neurons (SOMIs) in the dentate gyrus of mice as they run on a circular maze for rewards. Using silicon probes they record neural activity and optogenetically identify SOMIs. They find that in well-trained animals just over half of the SOMIs are activated prior to reward, suggesting they may be involved in reward anticipation. When the reward is shifted, the anticipatory licking at the previously rewarded location is reduced. In animals that did not show anticipatory licking, these responses were absent and SOMIs were active only after reward consumption. Finally, the authors inhibited SOMIs and found reduced flexibility in licking responses, suggesting that SOMIs may play a causal role in mediating flexible spatial behaviors.

Overall the manuscript is interesting, with a clear rationale for examining SOMIs during spatial exploration. There is substantial novelty in recording DG SOMIs with electrophysiology, characterizing them as local or projection, and directly inhibiting them. While there is some conceptual overlap with recent work (including from this group), the manuscript is still of broad interest as the specific role of SOMIs, particularly in DG, is still poorly understood. The experiments are generally well reasoned, technically sound, and rigorous. However, several weaknesses limit the interpretability of the data, but could be addressed with additional data.

We thank the reviewer for her/his appreciation and interest in our work, highlighting the currently limited information on the specific role of SOM-expressing interneurons in this brain area and the broad interest of this work for the neuroscience community.

Major Comments:

- The results from the chemogenetic inhibition experiment, as presented, do not clearly support the major premise of the manuscript, that DG SOMIs support flexible spatial behavior. The key metrics that are used to evaluate learning in this task (e.g., comparison of lick rates in the pre-reward zone to a baseline period, lick rates at the novel location) are not presented, and there appear to be changes in overall lick rates that make it difficult to interpret the results. There also appears to still be pre-emptive licking in the hm4Di group, which undermines the stated conclusions.*

Following the proposal of the reviewer, we revised **Figure 5** and included the baseline lick rate for control mice and animals expressing hM4Di in SOMIs. We observed no significant differences in the baseline lick rates between both groups ($P = 0.3607$, t -test; 4 and 5 mice respectively), which is now stated on **line 397** of the revised manuscript. In control mice, the lick rate in the original post-reward area went down to baseline levels indicating that animals recognized that the reward is no longer available at the trained location. The lick-rate in the post-reward area was therefore not different to the lick-rate in the novel pre-reward zone (revised **Figure 5e**, upper panel, post/blue vs pre/orange). However, the lick rate stayed high in the post-reward area of injected mice indicating that they did not recognize that the reward was no longer available. The lick-rate in the post-reward-area was therefore higher as compared to the pre-reward area in the novel reward zone (revised **Figure 5e**, lower panel, post/blue vs pre/orange). In summary, we conclude from these data that SOMI silencing in the DG causes a lacking updating of the novel information namely that rewards are no longer available at the originally train reward sites.

Additional Comments:

2. *Line 172 - "suggesting that licking per se had no apparent effect on SOMI firing." It still seems unclear whether licking is contributing to the SOMI response. While the licking rate outlasts the SOMI activation, the timing of lick initiation and SOMI firing is highly correlated and this certainly does not rule out that the two are related. In general, the use of anticipatory licking as a measure of behavior limits the experimental manipulations that could dissociate licking from performance.*

We removed this statement in the revised version of the manuscript. To further disentangle the potential contribution of the key behavioral variables licking, speed and reward delivery to SOMIs' activity, by first applying a general linear model (GLM; see also response to **Reviewer #2, major point 2a, b**). The model revealed a larger proportion of SOMIs for which the activity could be significantly explained by the behavioral parameters in E as compared to NE animals (Figure 2i). To further dissect the impact of the different variables in E mice, we followed the reviewers' advice and constructed reduced models in which one of the variables was randomly permuted. This analysis revealed a significant effect of licking and reward but not of running speed or acceleration. These data are now shown in the revised **Supplementary Figure 5c**. To further assess whether SOMI activity can be explained during anticipation of rewards, we ran the same analysis for a time window from 3 s before reward onset until reward onset. Similarly, only licking but not speed or acceleration significantly affected SOMI activity in E mice during the anticipatory window. These data are shown in the revised **Supplementary Figure 5d**. Thus, our data show that SOMI activity in experts at goal-locations is modulated by anticipatory behaviour. This set of novel data are presented on **page 6, lines 203-215**.

3. *A characterization of the viral expression and spread is necessary to interpret the hM4Di experiment*

We thank the reviewer for pointing to the missing data, which are now showing in the revised **Figure 5a**. The expression was restricted to the hilar DG and proximal CA3. In the DG, 89% of SOMIs identified by antibody labelling co-expressed hM4Di (5 mice). The specificity of the viral expression is stated in the **Figure 5a legend** and **line 682** of the revised manuscript.

4. *Many of the plots are heavily smoothed, and it appears they may even be smoothed across dimensions that should not be (e.g., across animals in Fig 2B or cells in Fig 3). Please clarify whether all smoothing is done only over time/space.*

The smoothing has only be applied over time and not across any other parameters. This is stated in the legend of **Figure 2** on **line 239** and the methods section on **line 757** of the revised manuscript.

4. *The authors focus on local and projection SOMIs but there are certainly more than 2 subtypes of SOMIs in DG hilus, which is clear from the variable responses observed in Fig 2E. Additional discussion of these, or perhaps a characterization of the firing properties of reward anticipatory versus non-responding SOMIs could be helpful.*

Our previous *in vitro* whole-cell recordings combined with intracellular labelling revealed two main SOMI types in the hilar DG based on their morphological and intrinsic physiological characteristics as well as local vs long-range connectivity (Yuan et al., 2017; eLife). We are therefore confident that based on those measures two main morphological and physiological subtypes are expressed within the DG, SOMI_{loc} and SOMI_{proj}. However, as stated by the reviewer, the responses of SOMIs at reward locations in expert mice was variable. The major fraction of all reward modulated SOMIs showed elevated (71%) whereas a minor fraction (29%) reduced activity at the expected pre-reward sites. As SOM cells in the DG are heavily connected with GCs and GABAergic cells, this variability is very likely caused by a network effect. The requested characterization of this variability is now stated in the results section on **page 6, lines 180-181** of the revised manuscript and we discuss this variability on **page 14, lines 472-474** of the revised manuscript.

5. *Since the reward pump is triggered ~1 sec before the reward is given (line 568), is it possible the SOMIs are responding to a conditioned auditory cue from the pump? The random sound cues presented in Supplemental Fig 6B partially addresses this, but it is still possible that a conditioned cue previously paired with reward could drive SOMI activity. (also see next comment).*

We measured the sound spectrum and sound intensity of the pump during reward delivery with a sound meter (Precision Instruments AL-1000), and detected a marked peak at ~11 kHz directly at the pump (green line in **Figure 6 for reviewers**). However, this peak was not detected at the running wheel for mice (red line). The distance between the pump and the recording site is ~1.5 m. Moreover, the pump sound spectrum and its intensity were similar to the values of the ambient noise in the room (red vs blue line). These data suggest that sound intensity declines over distance between the pump and the animals' position on the wheel and could therefore not be detected by the animal. We therefore assume that the pump sound is unlikely to act as a conditioned cue that paired with the reward could drive SOMI activity. This information is now added to the figure legend of **Supplementary Figure 7b (line 148-149)** and the revised Methods section on **page 18, lines 617-621**.

Figure 6 for Reviewers: The sound of the reward pump at ~11 kHz cannot be detected at the recording site. The intensity of the pump during reward delivery was plotted against sound frequency. The pump sound was directly recorded at the pump, in the same room where the experiments were conducted (green line), and at the running wheel for mice, located ~1.5 m distant from the wheel (red line). Data were compared with the frequency and intensity range of the ambient sounds in the room without the pump on (blue line). The arrow points to a peak at ~11 kHz near the pump, which is absent at the wheel. Each line is the average of 7 repeated measures. Note similar upper frequency range (>20 kHz) at the three recording conditions that mice preferentially hear.

7. *Supplemental 6B (Sound) - how can this example cell have a baseline firing rate around 150hz? It is dramatically different than all other cells. This does not seem physiological and undermines the conclusion of this panel. Please clarify this.*

We thank the reviewer for pointing to this exceptionally highly active cell. We checked again and can confirm that this cell was an optogenetically identified SOMI. Moreover, we confirm that the cell was highly discharging at the mentioned frequency. Under conditions of *in vitro* whole-cell recording we previously showed that during 1 sec positive current injections, hilar SOMIs reach a maximal average discharge frequency of 141.5 ± 5.7 Hz and can fire up to 210 Hz (Yuan et al., eLife 2017; their Figure 2J). As the cell in the Supplementary Figure 7B (sound) is the only SOM cell that we recorded with such a high discharge frequency, we decided to replace it with another more representative example cell.

8. *Additional details on how Experts and Non-Experts were split are needed in the results and methods sections.*

Following the request of the reviewer, we specified in more detail the two main properties that needed to be fulfilled to define an animal as expert or non-expert: licking frequency and running speed. In experts, licking frequency in the pre-reward area was significantly enhanced as compared to baseline values on the level of individual animals whereas in non-experts licking frequency remained constant. Moreover, running velocity was significantly reduced in the pre-reward zone as compared to the baseline period in experts but not in non-experts. This is now stated in the results section **on page 5, lines 161-163, 165-166** and the **Methods** section on **page 17, lines 571-578**. We further show in our **new Supplementary Figure 4f** of the revised manuscript that the performance score defined as percentage of laps during which increased lick rate was obtained in the pre-reward area compared to the remaining track was higher in expert mice than in non-experts with a clear distinction of experts at a performance score >55%.

9. *Line 304 - "To further proof that SOMI firing rapidly changes in ORI-zones during relocation" - This wording needs to be changed.*

Done. Now **lines 329-330**.

10. *Fig 5 - The viruses appear to be labeled incorrectly in this figure and caption. Please clarify if Control virus was also a FLEX virus.*

We corrected the labelling of the figure. We used an AAV-flex-hSyn-mCherry for controls, which has been **revised in Figure 5a**.

11. *Line 351 - "If DG SOMIs provide a confirmation signal of the expected outcome..." This was very confusing as this idea was not really introduced properly. Why do the authors hypothesize it is a confirmation signal?*

We apologize for the confusion and changed the sentence accordingly. The sentence can be found on **page 12, lines 383-384** of the revised manuscript.

12. *Fig 5D - what are the different colors within each subpanel? Please clarify*

We apologize for the missing information. Dark gray and pink refer to behavioral data obtained on day 1 whereas bright gray and pink refer to day 2. This information is now included in the **revised Figure 5d**.

13. *Could this "reward prediction" be driven by feedback input from overrepresentation of place cells near rewards? Is it possible to decouple place cell overrepresentation and this predictive coding? A discussion of this would be helpful.*

We thank the reviewer for pointing to the potential underlying network mechanisms that may cause predictive coding by SOMIs. In response to **major point 2c of Reviewer #2**, we performed additional data analysis of active putative granule cells (GCs) and mossy cells (MCs) within the DG of expert and non-expert mice. GCs form excitatory synapses onto SOMIs in the hilar area and we have previously shown in Yuan et al., eLife (2017) that these excitatory contacts can undergo associative long-term potentiation (LTP), which requires the timed generation of GC-mediated EPSPs preceding the action potential generation in the postsynaptic SOMI. Our single unit data show that there is an overlap in the rise of activity between GCs and SOMIs in the pre-reward zone (original Figure 2D orange trace compared to green traces in the **Figure 4D for reviewers**), suggesting that predictive coding by SOMIs might be influenced by presynaptic DG principal cell activity in experts. This is now discussed on **page 16, lines 545-553** of the revised manuscript.

Reviewer #4 (Remarks to the Author):

Point-by-point response NCOMMS-24-63761A

REVIEWERS' COMMENTS

Reviewer #1 (Remarks to the Author):

The authors have thoroughly addressed the critical comments and made significant revisions to clarify key sections and enhance the interpretation of the data. The manuscript has improved considerably as a result. I have no further suggestions.

We thank the reviewer for her/his positive evaluation of our revised version of the manuscript.

Reviewer #2 (Remarks to the Author):

The authors have addressed all the points I raised in my original revision. Crucially, they now provide extensive new additional statistical tests and controls that strongly support the specific modulation of SOMIs activity by reward expectation. They also show that this modulation is not present in non-expert mice and in other cell types such as fast-spiking interneurons. I have no further issues to raise and recommend this paper for publication. I believe it will be an important contribution to the field.

A small point regarding the characterization of SOMIs subtypes. The authors found no differences in some of their physiological properties (like modulation by LFP oscillations) and decided not to include them in the manuscript. I still believe this is useful information for future readers and worth including, but I leave it to the authors discretion on whether they prefer not to do it.

We thank the reviewer for his positive evaluation of our manuscript. We followed the advice of the reviewer and included the LFP oscillation data (previous Figure 3 for reviewers in the rebuttal letter) to **Supplementary Fig. 3** as **new panels (c & d)** with the corresponding figure legend and refer to this data on **page 4, lines 116-117** of the revised manuscript.

Reviewer #3 (Remarks to the Author):

The authors have done an excellent job of addressing my comments and those of the other reviewers. I think this is an extremely valuable contribution to our understanding of inhibitory control of flexible behaviors. I have only 1 minor comment: The revised manuscript includes rates of connectivity between mossy cells, granule cells, and SOMIs. The methods to differentiate these cell types was included in the author rebuttal, but should be included in the methods of the manuscript as well.

We thank the reviewer for her/his valuable comment and included the requested information (response 2c of reviewer #2) to the **Methods section** on **page 15 and 16, lines 577-583**.

Reviewer #4 (Remarks to the Author):
